# Modeling Collision-Coalescence in Particle Microphysics: Numerical Convergence of Mean and Variance of Precipitation in Cloud Simulations Using University of Warsaw Lagrangian Cloud Model (UWLCM) 2.1

Piotr Zmijewski, Piotr Dziekan, and Hanna Pawlowska

Institute of Geophysics, Faculty of Physics, University of Warsaw, Poland

**Correspondence:** Piotr Dziekan (pdziekan@fuw.edu.pl)

**Abstract.**

Numerical convergence of the collision-coalescence algorithm used in Lagrangian particle-based microphysics is studied in 2D simulations of an isolated Cumulus Congestus (CC) and in box/multi-box simulations of collision-coalescence. Parameters studied are the time step for coalescence and the number of super-droplets (SDs) per cell. Time step of $0.1\,\mathrm{s}$ gives converged droplet size distribution (DSD) in box simulations and converged mean precipitation in CC. Variances of the DSD and of precipitation are not sensitive to the time step. In box simulations, mean DSD converges for $10^3$ SDs per cell, but variance of the DSD does not converge as it decreases with increasing number of SDs. Fewer SDs per cell are required for convergence of the mean DSD in multi-box simulations, probably thanks to mixing of SDs between cells. In CC simulations, more SDs are needed for convergence than in box or multi-box simulations. Mean precipitation converges for $5 \times 10^3$ SDs, but only in a strongly precipitating cloud. In cases with little precipitation, mean precipitation does not converge even for $10^5$ SDs per cell. Variance in precipitation between independent CC runs is more sensitive to the resolved flow field than to the stochasticity in collision-coalescence of SDs, even when using as little as 50 SDs per cell.

## 1 Introduction

Particle microphysics (also known as Lagrangian particle-based microphysics, Lagrangian Cloud Model or super-droplet microphysics) is a class of Lagrangian methods for numerical modeling of cloud microphysics that has been developed in the last decade (Shima et al., 2009; Andrejczuk et al., 2010; Sölch and Kärcher, 2010; Riechelmann et al., 2012). In particle microphysics, numerical objects called super-droplets (also known as simulational particles) are used as proxies for hydrometeors. Similarly to the more common Eulerian bin models, particle models explicitly resolve evolution of the DSD. There are several advantages of particle models that make them a compelling alternative to bin models (Grabowski, 2020): lack of numerical diffusion, easy modeling of multiple hydrometeor attributes (e.g. chemical composition) and scaling down to direct numerical simulations, among others. However, modeling of collision-coalescence has proven to be difficult in particle microphysics. A few algorithms have been developed to do this, see Unterstrasser et al. (2017) for a review. Out of these, the all-or-nothing

(AON) algorithm from the Super-Droplet Method (SDM) of Shima et al. (2009) was shown to give the most accurate DSD in box simulations (Unterstrasser et al., 2017) and has been widely adopted (Hoffmann and Feingold, 2021; Dziekan et al., 2021; Unterstrasser et al., 2020; Arabas and Shima, 2013; Shima et al., 2020).

A recent study has found discrepancies in rain production between different particle models that use AON, although the models agree well in modeling condensational growth (Hill et al., 2023). This shows that better understanding of numerical convergence of AON is necessary before particle models can become the benchmark for microphysics modeling. Several studies have shown that AON is sensitive to the number of SDs, to the numerical time step and to the way SDs are initialized (Shima et al., 2009; Unterstrasser et al., 2017; Dziekan and Pawlowska, 2017; Schwenkel et al., 2018; Dziekan et al., 2019; Unterstrasser et al., 2020). Detailed studies of numerical convergence of AON were done so far only in simulations of pure collision-coalescence in a box (Unterstrasser et al., 2017) and in a 1D column (Unterstrasser et al., 2020). It was found that criteria for convergence of AON are different in 1D than in box simulations. This shows that to confidently model precipitation in LES with particle microphysics, it is not sufficient to use parameters that give convergence in a box or a 1D column. The aim of this study is to better understand numerical convergence of AON in LES.

We begin with a study of numerical convergence of AON in simulations of collision-coalescence in a box model. Although a similar study was done by Unterstrasser et al. (2017), we believe that it is valuable to repeat it for a number of reasons. Firstly, our implementation of particle microphysics differs in details from that of Unterstrasser et al. (2017), so it may converge differently. Understanding convergence of our implementation in a box model is useful for planning convergence tests in more realistic simulations done afterward. Secondly, we compare with one-to-one simulation (Dziekan and Pawlowska, 2017), which is a more detailed reference model than was used in Unterstrasser et al. (2017). This allows us to study convergence not only of the mean DSD, but also of the variance of the DSD. Lastly, we validate results of Unterstrasser et al. (2017).

Next, we study convergence of AON in multi-box simulations. This simulation type, introduced by Schwenkel et al. (2018), is similar to box simulations, but the domain is divided into multiple coalescence cells and super-droplets can move between these. In that way they are more similar to LES than box simulations and, unlike in LES, reference solutions can be produced by one-to-one simulations. The goal of multi-box simulations is to study how mixing of super-droplets between cells affects the minimum number of super-droplets required for convergence.

Finally, we study numerical convergence of AON in 2D simulations of isolated cumulus congestus. It is a much more realistic simulation than was used before for convergence tests of AON. The same processes are included as in a LES, with the only difference being smaller dimensionality. The reason why we use 2D instead of 3D is that this decreases the required computational power and memory size, allowing us to study a broader range of parameters. AON is a stochastic algorithm, so it gives different realizations of collision-coalescence in independent simulation runs. LES runs often also differ due to random differences in initial conditions. These differences in initial conditions include random perturbations of thermodynamic variables (e.g. temperature and humidity) and random initialization of SD attributes. Stochasticity in AON, as well as in initial conditions, leads to differences in flow fields, what can strongly impact results. To isolate the effect of stochasticity of AON from stochasticity of initial conditions, we use the same initial conditions in ensembles of simulations. Moreover, to facilitate studying convergence of AON, we use the same flow field for different simulations. This way, flow field is not affected by

different realizations of AON. We also perform reference "dynamic" simulations with differences in initial conditions and without a prescribed flow field. This allows us to assess the importance of stochasticity of AON relatively to other sources of stochasticity in LES.

We start with a presentation of the particle microphysics scheme, with emphasis on AON and on the SD initialization procedure (Section 2). Studies of numerical convergence of AON in box, multi-box and in 2D simulations are presented in Section 3, Section 4 and in Section 5, respectively. Conclusions for cloud modeling are discussed in Section 6.

## 2   Particle microphysics

In Lagrangian particle microphysics methods, particles in the air (aerosols, haze particles, cloud droplets, rain drops, ice particles) are represented by computational objects called super-droplets. In most cases, each SD represents numerous identical real particles. The number of real particles a SD represents is called its multiplicity $\xi$ (also known as weighting factor). Another commonly-used SD attribute is spatial position. Two additional attributes are useful for modeling warm microphysics: wet radius, which describes the total volume of a particle, and dry radius, which describes the volume of the dissolved matter. For most processes (advection, condensation, sedimentation), changes of SD attributes are described by the same equations that describe how single real particles are affected by these processes. However, it is not straightforward how to model collision-coalescence of SDs. In the next two sections, we present parts of the microphysics model that are particularly important for modeling collision-coalescence.

### 2.1   Initialization of SD radii and multiplicities

Multiplicities and radii of SDs are initialized from a prescribed initial size distribution. In this section, we describe common methods for doing this. The prescribed radius can either be wet or dry radius. We denote the initial number of SDs per grid cell with $N_{\mathrm{SD}}^{(\mathrm{init})}$.

In one initialization method, all SDs have the same multiplicities and their initial radii are drawn from the distribution using inverse sampling (Shima et al., 2009; Hoffmann et al., 2015). Multiplicity is equal to the initial number of droplets in a cell divided by $N_{\mathrm{SD}}^{(\mathrm{init})}$. Following Unterstrasser et al. (2017), we refer to this method as $\xi_{\mathrm{const}}$-init.

Another method of initialization is to divide the initial distribution into bins of equal sizes on a logarithmic scale. We denote the number of bins with $N_{\mathrm{SD}}^{(\mathrm{bin})}$. Within each bin we randomly select radius of a single SD, and its multiplicity follows from the initial distribution. It is not obvious what should be the choice of the leftmost and rightmost bin edges. Arabas et al. (2015) proposed to select bin edges so that multiplicities of SDs in the outermost bins is at least 1 (see Dziekan and Pawlowska (2017) for details of the algorithm). In this method, bin edges depend on the volume of grid cells and, more importantly, on $N_{\mathrm{SD}}^{(\mathrm{bin})}$. When $N_{\mathrm{SD}}^{(\mathrm{bin})}$ is increased, the largest possible initial SD radius is decreased. To counter this, Dziekan and Pawlowska (2017) proposed to initialize additional SDs using inverse sampling from part of the distribution to the right of the rightmost bin. Note that the number of these additional SDs is rather small. In all simulations presented in this paper,

we have $N_{\mathrm{SD}}^{(\mathrm{bin})} \leq N_{\mathrm{SD}}^{(\mathrm{init})} \leq 1.01\, N_{\mathrm{SD}}^{(\mathrm{bin})}$. Following Dziekan and Pawlowska (2017), we will refer to this method as "constant SD"-init.

Instead of using the algorithm for finding bin edges, one can simply prescribe them. We call this method "constant SD" fixed-init. In this method, no SDs are added to represent the part of the distribution to the right of the largest bin.

Unterstrasser et al. (2017) compared multiple methods of SD initialization and found that using bins to initialize radii (as in "constant SD"-init) is preferable because it requires the least SDs to correctly model collision-coalescence. In most of the simulations presented in this paper, we use the "constant SD"-init. In Section 5.6 we study sensitivity to SD initialization method.

## 2.2 Collision-coalescence of SDs: the AON algorithm

The AON algorithm, developed by Shima et al. (2009), is an algorithm for modeling collision-coalescence in Lagrangian particle microphysics. It is derived from the stochastic description of the collision-coalescence of particles (Gillespie, 1975). In this description, it is assumed that the probability of collision between a pair of particles is known. AON is designed to give the correct expected number of collisions and to keep the number of SDs constant. The drawback of AON is that it gives a variance in the number of collisions larger than the real variance (Shima et al., 2009). The probability that a pair of super-droplets $i$ and $j$ collide during some time interval is:

$$P_{ij}^{(\mathrm{s})} = \max(\xi_i, \xi_j) P_{ij}\,, \tag{1}$$

where $P_{ij}$ is the probability of collision between two real particles with identical attributes (e.g. wet radii) as SDs $i$ and $j$. It is given by $P_{ij} = K_{ij}\Delta t \Delta V^{-1}$, where $K_{ij}$ is the coalescence kernel, $\Delta t$ is the time interval and $\Delta V$ is cell volume (Shima et al., 2009). Coalescence of SDs $i$ and $j$ is defined as coalescence of $\min(\xi_i, \xi_j)$ pairs of real particles, each pair made of one particle represented by SD $i$ and one particle represented by SD $j$. Probability of SD coalescence can be greater than one, in particular for long time steps. This represents multiple coalescence of the SD pair (Shima et al., 2009). Multiple coalescence can be done only if the ratio of multiplicities of colliding SDs is sufficiently high. If it is not sufficiently high, then the number of coalescence events will be artificially decreased. Note that in $\xi_{\mathrm{const}}$-init it is not possible to have multiple coalescence between a SD pair, because all multiplicities are equal. For this reason, in our $\xi_{\mathrm{const}}$-init simulations we adapt time step for coalescence to maintain collision probability below 1. The "constant SD"-init method typically gives large differences between multiplicities of SDs. Thanks to that, multiple coalescence is usually possible, and we can use a constant time step for coalescence in this type of simulation.

In some implementations of AON, the number of super-droplet pairs tested for coalescence per time step is equal to $N_{\mathrm{SD}}(N_{\mathrm{SD}} - 1)/2$, where $N_{\mathrm{SD}}$ is the number of SDs in a coalescence cell (the coalescence cell is typically equivalent to an Eulerian grid cell, Dziekan and Pawlowska (2017)). This is known as quadratic sampling (Unterstrasser et al., 2017, 2020). However, the original AON method of Shima et al. (2009) uses a technique called linear sampling, which is designed to speed up the algorithm. In linear sampling, $\lfloor N_{\mathrm{SD}}/2 \rfloor$ non-overlapping pairs of super-droplets are considered per time step. The notation $\lfloor x \rfloor$ represents the largest integer less than or equal to $x$. To obtain the correct expected number of collisions in linear

sampling, the probability of collision between a pair of SDs is increased to:

$$P_{ij}^{(s,l)} = P_{ij}^{(s)} \frac{N_{SD}(N_{SD}-1)}{2} / \lfloor N_{SD}/2 \rfloor. \tag{2}$$

Linear and quadratic sampling techniques were directly compared in Dziekan and Pawlowska (2017) and in Unterstrasser et al.
(2020). Unterstrasser et al. (2020) showed that quadratic sampling converges for a longer time step than the linear sampling
(Figure 6 b therein). Once converged, both techniques give the same mean and variance (Dziekan and Pawlowska, 2017;
Unterstrasser et al., 2020). Typically, the number of collision pairs tested per unit of time is smaller in linear sampling than in
quadratic sampling, despite the shorter time steps. Moreover, in linear sampling all collision pairs can be computed in parallel
because they are non-overlapping. For these reasons, we use linear sampling in this work.

## 3 Box simulations

We model collision-coalescence of droplets in a well-mixed box. For simplicity, we use $r$ to denote wet radius in this section,
as the dry radius is not important for collision-coalescence. We analyze the mass density function $m(\ln r)$, which is such that
$m(\ln r) d\ln r$ is the mass of droplets per unit volume in the size range from $\ln r$ to $\ln r + d\ln r$. The initial distribution of $r$
is exponential in volume with $15\,\mu m$ mean wet radius and $142\,cm^{-3}$ droplet concentration, what gives $2\,g\,m^{-3}$ liquid water
content. This distribution was used in Onishi et al. (2015) and in Dziekan and Pawlowska (2017). The box volume is around
$0.45\,m^3$ and it initially contains 64 million droplets. Simulations are run for $300\,s$. We use a gravitational coalescence kernel
with collision efficiencies from Hall (1980) and from Davis (1972).

Three types of collision-coalescence models are compared: AON algorithm, one-to-one simulations and the stochastic coa-
lescence equation (SCE). AON is discussed in Section 2. One-to-one simulations are particle simulations with $\xi = 1$, i.e. each
real droplet is explicitly modeled. We use $\xi_{const}$-init and linear sampling in one-to-one simulations. One-to-one simulations
produce a realization in agreement with the master equation (Dziekan and Pawlowska, 2017). As such, they are the most fun-
damental type of simulation used and are considered to produce reference results. SCE is an equation for time evolution of
the average DSD. It is typically used to model collision-coalescence in bin models. Dziekan and Pawlowska (2017) showed
that the SCE gives correct average results for droplet populations greater than $10^7$, so it should be valid in the box simulation
with $6.4 \times 10^7$ droplets discussed in this paper. We solve SCE with the Bott (1997) flux method with bin scaling parameter
$\alpha = 2^{1/10}$ and time step $0.1\,s$. These parameters were found to give converged results.

One-to-one and AON simulations are stochastic. We run an ensemble of one-to-one simulations and ensembles of AON
simulations for different values of $N_{SD}^{(bin)}$. From these, we calculate the ensemble mean of the mass density function $\langle m \rangle$ and
the ensemble standard deviation of the mass density function $\sigma(m)$. The ensemble size in one-to-one simulations is $\Omega = 10$.
In AON simulations, it is $\Omega = 10^6/N_{SD}^{(bin)}$ for $N_{SD}^{(bin)} \leq 100$ and $\Omega = 10^7/N_{SD}^{(bin)}$ for $N_{SD}^{(bin)} > 100$. The SCE is deterministic,
therefore it does not require a simulation ensemble and it does not explicitly model variance of the DSD. However, Gillespie
(1975) estimated the variance of the number of droplets in a given size range to be equal to the number of droplets in this size
range. We validate this estimate by comparing it with one-to-one results.

## 3.1 Results of box simulations

First, we check how numerical time step $\Delta t_{\text{coal}}$ affects AON simulations with $N_{\text{SD}}^{(\text{bin})} = 10^2$, which is the number of SDs typical for LES. In Fig. 1, we show DSDs at the end of the simulation for different time step lengths. There are no differences in $\langle m \rangle$ between $\Delta t_{\text{coal}} = 0.1\,\text{s}$ and $\Delta t_{\text{coal}} = 0.01\,\text{s}$. Using $\Delta t_{\text{coal}} = 1\,\text{s}$ results in too large $\langle m \rangle$ for the largest droplets. $\Delta t_{\text{coal}} = 10\,\text{s}$ gives yet larger $\langle m \rangle$ for the largest droplets and also a decrease in $\langle m \rangle$ for droplets with radii between $40\,\mu\text{m}$ and $100\,\mu\text{m}$. Regarding fluctuations, we see that differences in $\sigma(m)$ correspond to differences in $\langle m \rangle$, e.g. too large $\langle m \rangle$ for largest droplets also gives too large $\sigma(m)$ for largest droplets (Fig. 1 (a) and (b) for $\Delta t_{\text{coal}} = 10$). From this test, we conclude that mean DSD converges for $\Delta t_{\text{coal}} = 0.1\,\text{s}$ and that fluctuations in DSD are not sensitive to $\Delta t_{\text{coal}}$.

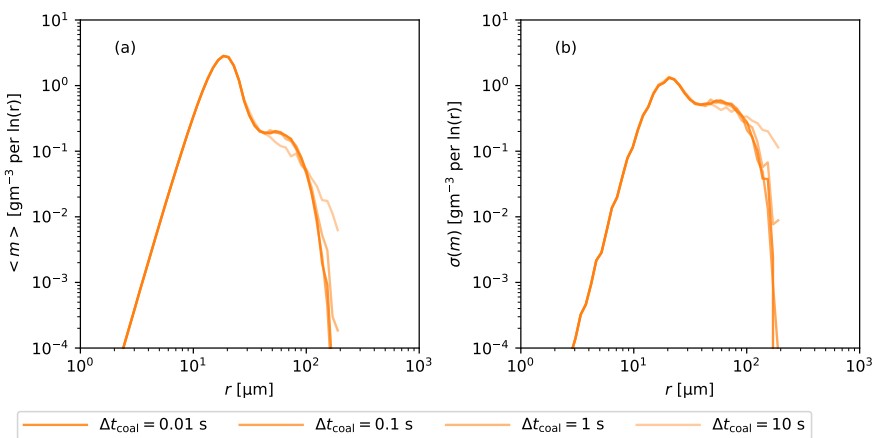

**Figure 1.** (a) Mean and (b) standard deviation of the mass density function $m$ for $N_{\text{SD}}^{(\text{bin})} = 10^2$ box simulations at $t = 300\,\text{s}$ for different time step lengths.

Next we check how results are affected by the number of SDs for $\Delta t_{\text{coal}} = 0.1\,\text{s}$. In Fig. 2 we show DSDs at the beginning and at the end of an AON simulation for different $N_{\text{SD}}^{(\text{bin})}$. Results of one-to-one simulations and of the SCE are plotted for reference. The initial $\langle m \rangle$ is very well represented for all methods of radius initialization and agrees well with the SCE initialization (Fig. 2 (a)). The first 11 moments of the initial distribution are plotted in the Supplement. The difference between moments from simulations and the expected moments is up to $11\,\%$ in AON and up to $3\,\%$ in SCE. The difference between moments in AON for various $N_{\text{SD}}^{(\text{bin})}$ is negligible. Differences in initialization may cause differences between results of AON and SCE, but not between results of AON with different $N_{\text{SD}}^{(\text{bin})}$. The initial $\sigma(m)$ decreases approximately linearly with increasing $N_{\text{SD}}^{(\text{bin})}$ (Fig. 2 b). $N_{\text{SD}}^{(\text{bin})} = 10^5$ gives smaller initial $\sigma(m)$ than one-to-one, despite much higher number of SDs in the latter ($6.4 \times 10^7$). The reason for this are the differences in the radius initialization procedure. The mean DSD at the end of the simulation does not significantly differ between different types of simulations, except for $N_{\text{SD}}^{(\text{bin})} = 10$, which gives too little droplets with radii between $30\,\mu\text{m}$ and $130\,\mu\text{m}$, and too many droplets with $r > 130\,\mu\text{m}$ (Fig. 2 c). Fluctuations in DSD at the end of AON simulations decrease with increasing $N_{\text{SD}}^{(\text{bin})}$ (Fig. 2 d). We find that standard deviation $\sigma(m)$ is

approximately proportional to $\sqrt{N_{\text{SD}}^{(\text{bin})}}^{-1}$, in particular for $10^3 \leq N_{\text{SD}}^{(\text{bin})} \leq 10^5$. Since multiplicity is inversely proportional to $N_{\text{SD}}^{(\text{bin})}$, standard deviation is proportional to the square root of multiplicity, as estimated by Shima et al. (2009). Even for $N_{\text{SD}}^{(\text{bin})} = 10^5$, $\sigma(m)$ in AON is much larger than the reference one-to-one result. It is seen that $\sigma(m)$ estimated from the SCE as the square root of the number of droplets, as proposed by Gillespie (1975), is larger than the one-to-one result, but much closer to it than $\sigma(m)$ in AON with $N_{\text{SD}}^{(\text{bin})} = 10^5$.

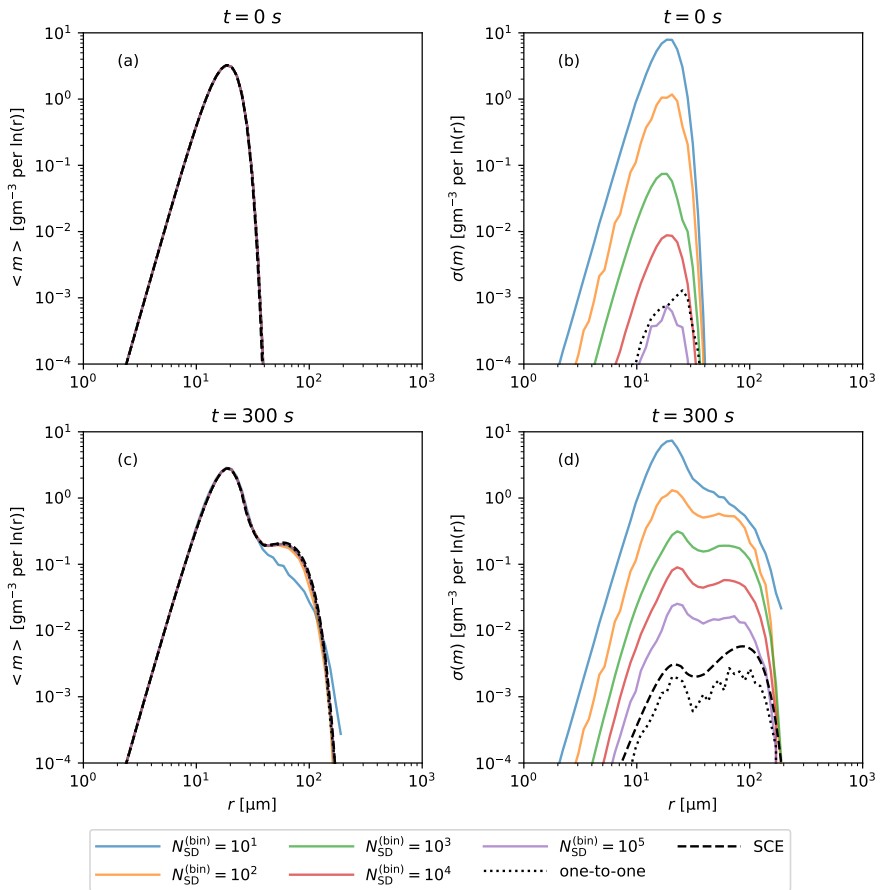

**Figure 2.** Mean and standard deviation of the mass density function $m$ from box simulations at $t = 0\,$s (a-b) and at $t = 300\,$s (c-d). Two types of Lagrangian simulations (SDM with $\Delta t_{\text{coal}} = 0.1\,$s and with various $N_{\text{SD}}^{(\text{bin})}$; one-to-one simulations) and solutions to the SCE are compared.

Differences in $\langle m \rangle$ between simulations for different combinations of $N_{\text{SD}}^{(\text{bin})}$ and of $\Delta t_{\text{coal}}$ may potentially lead to differences in the mean amount of rain in LES with Lagrangian particle microphysics. To have a better view of this issue, we plot differences between $\langle m \rangle$ in one-to-one simulations and $\langle m \rangle$ in AON simulations with different $N_{\text{SD}}^{(\text{bin})}$ (Fig. 3). In the plot, $\langle m \rangle$ is multiplied by the terminal velocity to get the sedimentation mass flux of droplets of given size. This analysis

confirms that results converge for $\Delta t_{\text{coal}} = 0.1\,\text{s}$, irrespective of $N_{\text{SD}}^{(\text{bin})}$ (Fig. 3 a-c). Longer time steps result in underestimation of the sedimentation flux for droplets with radii between around 40 and around 120 microns, and in overestimation of sedimentation flux for other droplets. Regarding convergence with $N_{\text{SD}}^{(\text{bin})}$, we find that AON results agree with one-to-one for $N_{\text{SD}}^{(\text{bin})} \geq 10^3$ (Fig. 3 d). For $N_{\text{SD}}^{(\text{bin})} = 10$, the mass flux is overestimated for $r < 30\,\mu\text{m}$ and $r > 130\,\mu\text{m}$ and underestimated for $30\,\mu\text{m} < r < 130\,\mu\text{m}$. Using $N_{\text{SD}}^{(\text{bin})} = 10^2$ underestimates mass flux for $r > 50\,\mu\text{m}$ (Fig. 3 d).

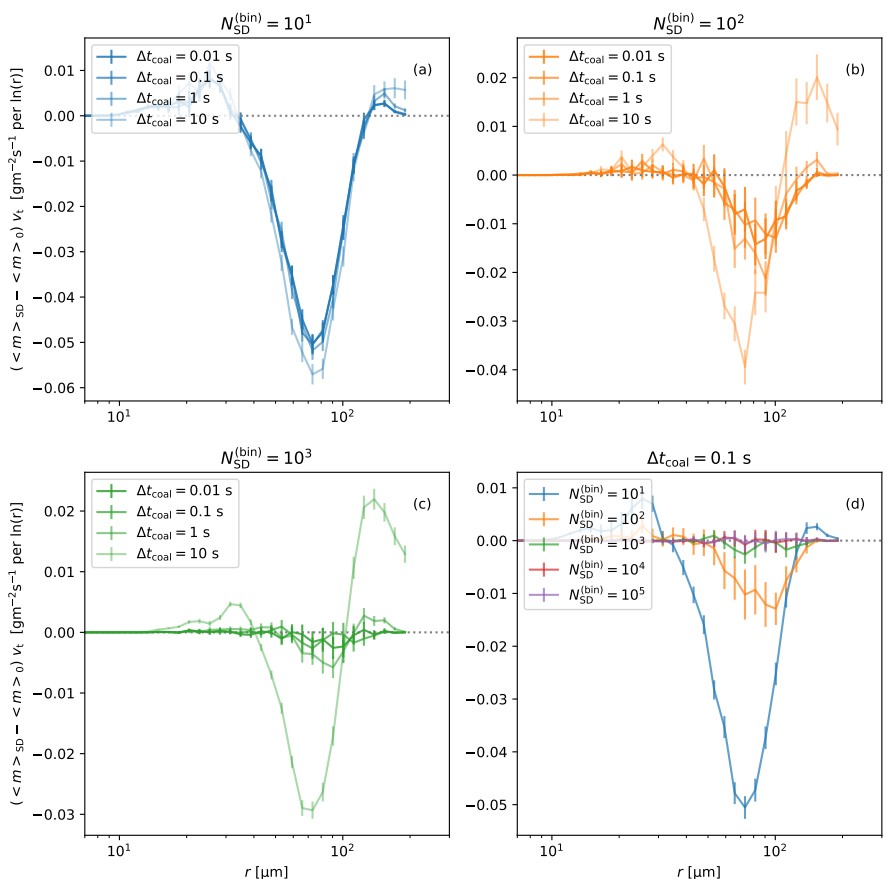

**Figure 3.** Differences between $\langle m \rangle$ from SDM ($\langle m \rangle_{\text{SD}}$) and one-to-one ($\langle m \rangle_0$) Lagrangian microphysics, multiplied by terminal velocity $v_t$. Results of box model simulations at $t = 300\,\text{s}$. Vertical error bars show the $95\,\%$ confidence interval.

Initialization of droplet radii in Lagrangian particle microphysics is often stochastic (Shima et al., 2009; Unterstrasser et al., 2017; Dziekan and Pawlowska, 2017). It is a source of random differences between simulations that is separate from the stochastic collision-coalescence algorithm. We want to check how important are these two different sources of randomness for variance in the modeled DSD. We run ensembles of simulations that do not differ in the initial DSD, but differ only in the realization of collision-coalescence. Comparison of these simulations with simulations that differ both in the initial DSD and the realization of collision-coalescence is shown in Fig. 4. The comparison is done for one-to-one simulations and for AON

simulations with $N_{\mathrm{SD}}^{(\mathrm{bin})} = 100$. The initial $\langle m \rangle$ agrees well for all types of simulations (Fig. 4 a). As expected, the initial $\sigma(m)$

is equal to zero for simulations without randomness in the initial DSD (Fig. 4 b). At the end of the simulation, $\langle m \rangle$ agrees well between simulations with and without randomness in initial DSD (Fig. 4 c). For droplets with $r > 20\,\mu\mathrm{m}$, $\sigma(m)$ at the end of the simulation is also not sensitive to randomness in the initial DSD (Fig. 4 d). Lack of randomness in the initial DSD results in slightly smaller $\sigma(m)$ for $r < 20\,\mu\mathrm{m}$, $\sigma(m)$ (Fig. 4 d). Considering that collision-coalescence is responsible for formation of large droplets, and that smaller droplets are formed by condensation, we conclude that the randomness in the initial DSD is

not important for mean nor fluctuations in large droplet production.

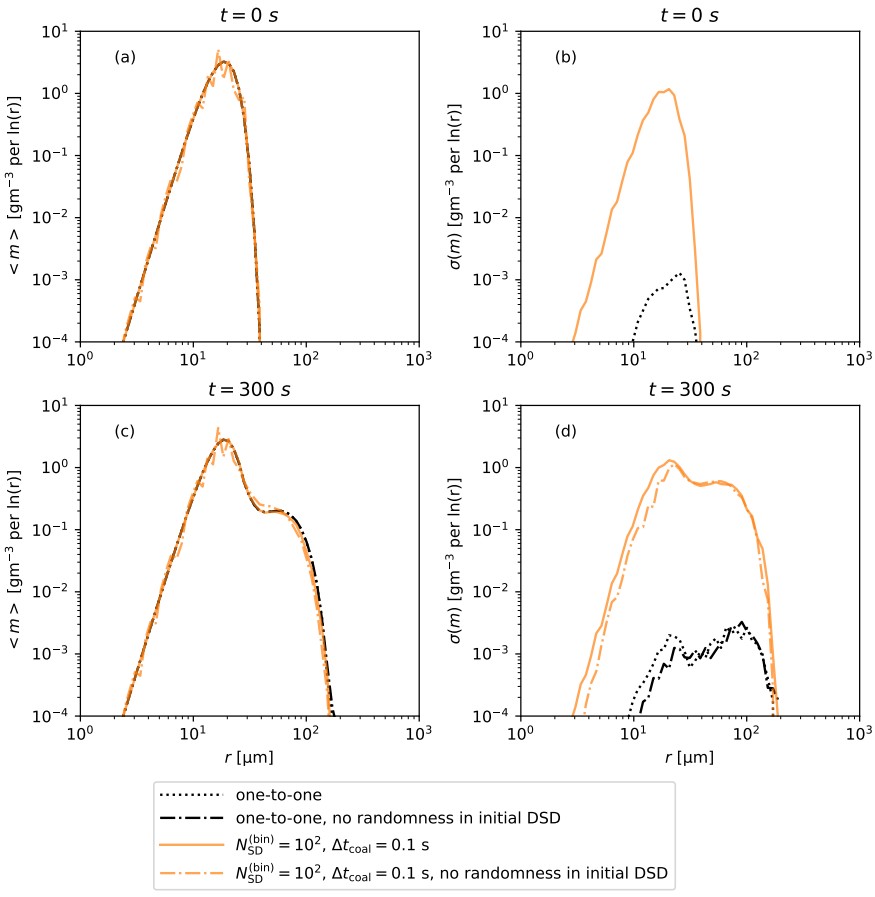

**Figure 4.** As in Fig. 2, but comparing ensembles of simulations with and without randomness in the initial DSD.

## 3.2 Summary of box simulations and comparison with previous studies

Box simulations of collision-coalescence with AON show convergence of $\langle m \rangle$ for $\Delta t_{\mathrm{coal}} \leq 0.1\,\mathrm{s}$, irrespective of $N_{\mathrm{SD}}^{(\mathrm{bin})}$, and for $N_{\mathrm{SD}}^{(\mathrm{bin})} \geq 10^3$. The standard deviation $\sigma(m)$ is not sensitive to $\Delta t_{\mathrm{coal}}$, but decreases with increasing $N_{\mathrm{SD}}^{(\mathrm{bin})}$. In one-to-one

simulations, variance of the number of droplets in a size bin is approximately equal to the number of droplets in the bin, as predicted by Gillespie (1975). This relationship could be used to model the stochastic nature of collision-coalescence in bin microphysics.

Box model tests of the mean DSD in the AON algorithm were previously done by Shima et al. (2009) and by Unterstrasser et al. (2017). Unterstrasser et al. (2020) did column simulations and some of them did not include sedimentation, which is equivalent to box simulations. Shima et al. (2009) found relatively good agreement with the SCE for $\Delta t_{\mathrm{coal}} = 0.1\,\mathrm{s}$ and for $N_{\mathrm{SD}}^{(\mathrm{bin})} \approx 2 \times 10^6$. The time step requirement is the same as found in this work, but the required $N_{\mathrm{SD}}^{(\mathrm{bin})}$ is much higher. The latter is probably because Shima et al. (2009) used the constant multiplicity initialization, which was found by Unterstrasser et al. (2017) to require many more SDs than their "singleSIP" initialization, which is similar to our constant SD initialization.

Using the "singleSIP" initialization, Unterstrasser et al. (2017) showed that results are close to converging for $\Delta t_{\mathrm{coal}} = 1\,\mathrm{s}$ (Fig. 18 therein, first column). An important difference between this work and Unterstrasser et al. (2017) is that the latter used quadratic sampling. Differences between linear and quadratic sampling methods are discussed in Section 2.2. Regarding convergence with $N_{\mathrm{SD}}^{(\mathrm{bin})}$, Unterstrasser et al. (2017) found convergence for $N_{\mathrm{SD}}^{(\mathrm{bin})} \geq 10^3$ (Fig. 18 therein, second column; there $\kappa = 200$ corresponds to $N_{\mathrm{SD}}^{(\mathrm{bin})} \approx 10^3$) and Unterstrasser et al. (2020) did not find convergence for up to $N_{\mathrm{SD}}^{(\mathrm{bin})} = 10^3$ (Fig. 6 a therein). Convergence tests in Unterstrasser et al. (2017) and in Unterstrasser et al. (2020) were done by analyzing moments of the DSD, and it was most difficult to obtain convergence of the 0-th moment (total droplet number). We find contrary results, i.e. that higher moments converge more slowly than lower moments. To illustrate this, we consider our box simulations for $N_{\mathrm{SD}}^{(\mathrm{bin})} = 10^2$. In these simulations, $\Delta t_{\mathrm{coal}} = 10\,\mathrm{s}$ gives visibly different large end of the DSD than $\Delta t_{\mathrm{coal}} = 0.1\,\mathrm{s}$ (Fig. 1). The 0-th (total droplet number) and 2-nd (radar reflectivity) moments are larger for $\Delta t_{\mathrm{coal}} = 10\,\mathrm{s}$ than for $\Delta t_{\mathrm{coal}} = 0.1\,\mathrm{s}$ by approximately $0.5\,\%$ and $30\,\%$, respectively. This is consistent with the intuition that higher moments are more sensitive to the large end of the DSD, and it is the large end of the DSD that is most sensitive to collision-coalescence.

The AON implementation from the *libcloudph++* library, which is used in this paper, was also used in box simulations described in Dziekan and Pawlowska (2017). That paper discussed convergence of $t_{10\%}$, the time after which $10\,\%$ of cloud mass is turned into rain mass. Dziekan and Pawlowska (2017) found that for $\Delta t_{\mathrm{coal}} = 1\,\mathrm{s}$ mean $t_{10\%}$ converges for $N_{\mathrm{SD}}^{(\mathrm{bin})} \geq 10^3$ (Fig. 4 therein). Dziekan and Pawlowska (2017) also showed that the standard deviation of $t_{10\%}$ decreases linearly with the square root of $N_{\mathrm{SD}}^{(\mathrm{bin})}$ (Fig. 5 therein). This is in agreement with our observation that $\sigma(m)$ is proportional to $\sqrt{N_{\mathrm{SD}}^{(\mathrm{bin})}}^{-1}$ and with the theoretical prediction from Shima et al. (2009) (Sec. 4.1.4 therein).

## 4 Multi-box simulations

Simulation setup is the same as in box simulations, but the domain is divided into $C$ equal rectangular cells of volume $0.45\,\mathrm{m}^3/C$. Only SDs that are in the same cell can collide with each other. Super-droplets move around the domain with a velocity that is a sum of the terminal velocity and of the air velocity, which is calculated from a synthetic isotropic turbulence model. Details of the turbulence model are given in Appendix A. Side walls are periodic. The turbulent kinetic energy dissipation rate is $10\,\mathrm{cm}^2\mathrm{s}^{-3}$. Simulations are run for $300\,\mathrm{s}$. The model time step is adapted to keep the Courant number in each

direction smaller than one (but the time step is not longer than $0.1\,\mathrm{s}$). The coalescence time step is equal to the model time step. When initializing SDs, the entire domain is treated as a single cell. Therefore, $N_{\mathrm{SD}}^{(\mathrm{bin})}$ represents the total initial number of SDs (minus SDs from 'tail' initialization). The mean number of SDs per cell at the start of the simulation is $N_{\mathrm{SD}}^{(\mathrm{cell})} = N_{\mathrm{SD}}^{(\mathrm{bin})}/C$.

Initial SD positions are selected randomly within the domain.

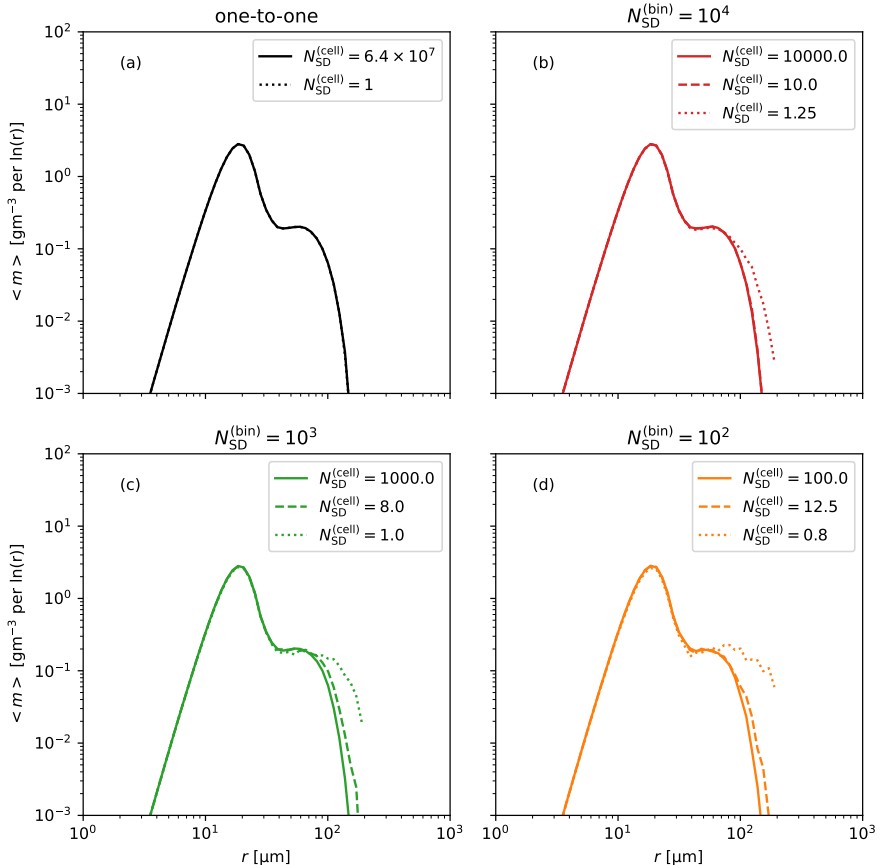

**Figure 5.** Mean mass density function at the end of multi-box simulations from (a) one-to-one simulations and AON simulations with $N_{\mathrm{SD}}^{(\mathrm{bin})} = 10^4$ in (b), with $N_{\mathrm{SD}}^{(\mathrm{bin})} = 10^3$ in (c) and with $N_{\mathrm{SD}}^{(\mathrm{bin})} = 10^2$ in (d). Each figure shows results for differing number of cells, corresponding to differing $N_{\mathrm{SD}}^{(\mathrm{cell})}$.

We run ensembles of one-to-one and AON simulations for differing number of cells. The ensemble size is $\Omega = 30$ in one-to-one simulations and $\Omega = 3 \times 10^6 / N_{\mathrm{SD}}^{(\mathrm{bin})}$ in AON simulations. The DSD at the end of the simulation is shown in Fig. 5. In principle, one-to-one simulations should become more realistic as the number of cells is increased ($N_{\mathrm{SD}}^{(\mathrm{cell})}$ is decreased). This is because droplet collisions are local, i.e. droplets need to get close together in order to collide. We find that in one-

to-one simulations, there is no difference in results for $N_{\mathrm{SD}}^{(\mathrm{cell})} = 1$ and for $N_{\mathrm{SD}}^{(\mathrm{cell})} = 6.4 \times 10^7$ (Fig. 5 (a)). This shows that there is no error in assuming that the domain is well-mixed, i.e. that droplets are uniformly distributed within the domain.

In AON simulations, each SD represents multiple droplets that are uniformly distributed within the cell in which the SD is located. As $C$ is increased while $N_{\mathrm{SD}}^{(\mathrm{bin})}$ is kept constant, all droplets represented by a SD are confined to a smaller volume. Therefore, increasing $C$ leads to less uniform spatial distribution of droplet sizes. Results show that this can cause errors in the domain-averaged DSD (Fig. 5 (b)-(d)). For $N_{\mathrm{SD}}^{(\mathrm{bin})}$ large enough, box simulations ($C = 1$, $N_{\mathrm{SD}}^{(\mathrm{cell})} = N_{\mathrm{SD}}^{(\mathrm{bin})}$) agree with one-to-one simulations (Fig. 3 (d)). However, the same $N_{\mathrm{SD}}^{(\mathrm{bin})}$ but with large $C$ (small $N_{\mathrm{SD}}^{(\mathrm{cell})}$) results in production of too large droplets (Fig. 5 (b)-(d)). The minimal value of $N_{\mathrm{SD}}^{(\mathrm{cell})}$ that gives correct results depends on $N_{\mathrm{SD}}^{(\mathrm{bin})}$: $N_{\mathrm{SD}}^{(\mathrm{cell})}$ can be smaller for larger $N_{\mathrm{SD}}^{(\mathrm{bin})}$ (e.g. $N_{\mathrm{SD}}^{(\mathrm{cell})} = 10$ works well for $N_{\mathrm{SD}}^{(\mathrm{bin})} = 10^4$, but $N_{\mathrm{SD}}^{(\mathrm{cell})} = 12.5$ gives errors for $N_{\mathrm{SD}}^{(\mathrm{bin})} = 10^2$). The number of coalescence cells does not affect $\sigma(m)$ (not shown in the figure).

In line with conclusions of Schwenkel et al. (2018) and of Unterstrasser et al. (2020), multi-box simulations show that fewer SDs per cell are needed to correctly model collision-coalescence when mixing of SDs between cells is included. For example, box simulations with $N_{\mathrm{SD}}^{(\mathrm{bin})} = N_{\mathrm{SD}}^{(\mathrm{cell})} = 10$ give significant errors, but multi-box simulations with $N_{\mathrm{SD}}^{(\mathrm{bin})} = 10^4$ and $N_{\mathrm{SD}}^{(\mathrm{cell})} = 10$ are close to reference. It is important that the rate of intercell mixing decreases with increasing cell size, which affects the minimal required $N_{\mathrm{SD}}^{(\mathrm{cell})}$. Simulations for $N_{\mathrm{SD}}^{(\mathrm{bin})} = 10^2$ and $N_{\mathrm{SD}}^{(\mathrm{cell})} = 12.5$ give errors, while simulations for $N_{\mathrm{SD}}^{(\mathrm{bin})} = 10^4$ and $N_{\mathrm{SD}}^{(\mathrm{cell})} = 10$ give correct results. Cells are larger in the former case than in the latter case. Larger cells imply less intecell mixing and, in consequence, larger $N_{\mathrm{SD}}^{(\mathrm{cell})}$ required for convergence.

## 5   2D Cumulus Congestus Simulations

In this section, we analyze AON in a two-dimensional simulation of an isolated cumulus congestus cloud. Conclusions about convergence of AON in box and multi-box simulations that were presented in the previous sections do not necessarily apply to higher dimensional simulations or simulations that include more processes affecting the DSD (e.g. condensational growth). For example, Unterstrasser et al. (2020) found that it is easier to reach convergence in a one-dimensional column simulation than in a box simulation. Based on the fact that in box simulations $\langle m \rangle$ converges for large $N_{\mathrm{SD}}^{(\mathrm{bin})}$, we can expect that precipitation in the CC simulation also converges for large $N_{\mathrm{SD}}^{(\mathrm{bin})}$. However, it is possible that precipitation in CC is affected by the artificially large variance in AON, which is illustrated by the lack of convergence of $\sigma(m)$ in box simulations. Too large variance in AON may result in too large differences in the DSD between cells. This might affect precipitation averaged over the entire cloud, because there is mixing between cells. Too large variance may also cause too large differences in precipitation between independent simulation runs. However, it is possible that in cloud simulations spatial and temporal variability of DSD is more susceptible to other factors, e.g. changes in relative humidity. Then, too large variability in AON would not be a problem. Variance in the number of collisions is inversely proportional to $N_{\mathrm{SD}}^{(\mathrm{bin})}$ (see Section 3). Doing CC simulations for different values of $N_{\mathrm{SD}}^{(\mathrm{bin})}$ allows us to study how the artificially large variance in AON affects simulations, even though it is not possible to have $N_{\mathrm{SD}}^{(\mathrm{bin})}$ large enough for the variance to converge. Box simulations suggest that the time step typically used in LES is sufficient for convergence of AON, but we also do a time step convergence test in CC.

## 5.1 LES model and setup

The CC simulations are done with the University of Warsaw Lagrangian Cloud Model (UWLCM). UWLCM is a LES tool that allows 2D and 3D simulations with Lagrangian particle (or Eulerian bulk) microphysics. Thermodynamic variables (potential temperature, water vapor mixing ratio, velocity) are modeled in an Eulerian manner. The Lipps-Hemler anelastic approximation (Lipps and Hemler, 1982) is used to filter acoustic waves. For spatial discretization of Eulerian variables, the staggered Arakawa-C grid (Arakawa and Lamb, 1977) is used. The finite-difference method is used to solve equations for Eulerian variables. The multidimensional positive-definite advection transport algorithm (MPDATA) (Smolarkiewicz, 2006) is used to model transport of Eulerian variables. The model uses the generalized conjugate residual solver (Smolarkiewicz and Margolin, 2000) to solve the pressure disturbance. In this paper, sub-grid scale (SGS) transport of Eulerian variables is modeled using the implicit LES approach (Grinstein et al., 2007). Depending on the simulation type, SGS advection of SDs is either ignored or modeled as an Ornstein-Uhlenbeck (hereby OU) process that crudely represents homogeneous isotropic turbulence (eq. (10) in Grabowski and Abade (2017)). A more detailed description of UWLCM can be found in Dziekan et al. (2019) and in Dziekan and Zmijewski (2022).

We use an isolated cumulus congestus modeling setup that was one of the cases studied at the International Cloud Modeling Workshop 2020. It is an adaptation of the setup developed by Lasher-Trapp et al. (2001). The computational domain is 12 km in horizontal and 10 km in vertical. Vertical profiles come from a conditionally unstable sounding from the Small Cumulus Microphysics Study field campaign. Initial potential temperature and water vapor mixing ratio fields are randomly perturbed below $1 \, \text{km}$ altitude. Perturbation amplitudes are $0.025 \, \text{g} \, \text{kg}^{-1}$ and $0.01 \, \text{K}$. For the first hour, surface fluxes are uniform: $0.04 \, \text{g} \, \text{kg}^{-1} \, \text{m} \, \text{s}^{-1}$ latent heat flux and $0.1 \, \text{K} \, \text{m} \, \text{s}^{-1}$ sensible heat flux. Afterward, surface fluxes have a Gaussian distribution centered at the middle of the domain, with maxima three times larger than the uniform flux from the first hour and with half width of $1.7 \, \text{km}$. The momentum surface flux is given by a constant friction velocity $0.28 \, \text{m} \, \text{s}^{-1}$. The total simulation time is 3 hours. The lateral boundaries are periodic, and the upper boundary is free-slip rigid-lid. We use an aerosol distribution based on observations from the RICO campaign (VanZanten et al., 2011). The distribution is made of two log-normal modes. The first (second) mode parameters are: number concentration $90 \, \text{cm}^{-3}$ ($15 \, \text{cm}^{-3}$), geometric mean radius $0.03 \, \mu\text{m}$ ($0.14 \, \mu\text{m}$) and geometric standard deviation $1.28$ ($1.75$). Aerosol type is ammonium bisulfate. We model these relatively clean conditions in order to have significant amount of precipitation, which is the focus of this study. A gravitational coalescence kernel is used, with collision efficiencies from Hall (1980) for large droplets and from Davis (1972) for small droplets. The coalescence efficiency is set to one. There is no droplet breakup. Terminal velocities are calculated using a formula of Khvorostyanov and Curry (2002). Model time step is $0.5 \, \text{s}$, time step for condensation is $0.1 \, \text{s}$ and cell size is $100 \, \text{m}$ in each direction.

We use 2D instead of 3D LES, because it allows us to study much larger values of $N_{\text{SD}}^{(\text{bin})}$. The same processes are modeled in 2D as in 3D, e.g. condensation, advection, sedimentation, collision-coalescence, etc. In 2D, the modeled flow field has different characteristics than in 3D. We expect to see more variability between simulation runs in 2D than there would be in 3D, because of a much smaller number of spatial cells. Rate of mixing of SDs between cells can also be different in 2D than in 3D. However,

we think that the way this variability is affected by parameters of the microphysics scheme in 2D is representative of how it would be affected in 3D.

## 5.2 Simulation strategy

Typically, in LES there is a random perturbation of initial conditions, e.g. of temperature and humidity. In LES with particle microphysics, initial conditions may also differ in SD attributes, because they are often randomly initialized. This randomness in initial conditions leads to differences in results between simulation runs, independently of AON. To understand the role of AON, we isolate its effect by comparing dynamic and kinematic simulations. In dynamic simulations, the pressure equation is solved, meaning that different realizations of microphysics lead to different flow fields. In kinematic simulations, flow field is prescribed. Our strategy is to run an ensemble of dynamic simulations, denoted with D, with random differences in initial conditions. We consider this ensemble as a control group, because this is the way LES is usually done. From dynamic simulations, we select three realizations: one with little, one with medium and one with high amount of rain (LR, MR and HR, respectively). Flow fields from these simulations are used to run ensembles of kinematic simulations. In kinematic simulations, initial conditions do not change within an ensemble. Therefore, any variability within a kinematic ensemble is solely caused by AON. Our goal is to study convergence of precipitation, which is a variable sensitive to modeling collision-coalescence. We also study convergence in simulations without collision-coalescence, to make sure that convergence in simulations with collision-coalescence is only related to AON. Number of simulations of each type is give in Table 2.

To generate velocity fields for kinematic simulations, we run numerous dynamic simulations. Our goal is to find three velocity fields that would give significantly different amounts of rain. In a single dynamic simulation, the amount of precipitation depends not only on the realized flow field, but also on the realization of the AON algorithm. This means that rain from a single dynamic simulation is not representative of the expected amount of rain from a series of simulations with the same velocity field. To be sure that we select velocity fields that will give different amounts of rain, first we chose a candidate velocity fields based on the amount of rain in the single dynamic run, and then we ran 20 kinematic simulations and used the average from these 20 simulations as the expected amount of rain for a given velocity field. Note that these 20 simulations were just a preliminary ensemble to estimate the expected amount of precipitation, and that the final number of simulations was much larger (it is given in Table 2). Based on this procedure, we selected the three velocity fields for kinematic simulations: LR, MR and HR. The histogram of the distribution of accumulated surface precipitation in the dynamic simulation ensemble is shown in Fig. 6. These highlighted bins delineate the range of original precipitation values from the velocity fields that were considered.

Besides using different flow fields, we study sensitivity to the model of SGS advection of SDs, to the SD initialization method, and we run simulations without collision-coalescence. A list of all simulation types is given in Table 1.

## 5.3 Temporal development of cloud

In this section we discuss time series of general cloud properties in the D, LR, MR and HR scenarios (with collision-coalescence). This is done to give the readers an idea about how the modeled cloud develops. Time series of cloud top height (CTH), cloud cover (cc), cloud water path (CWP), rain water path (RWP) and surface precipitation are plotted in Fig. 7. The

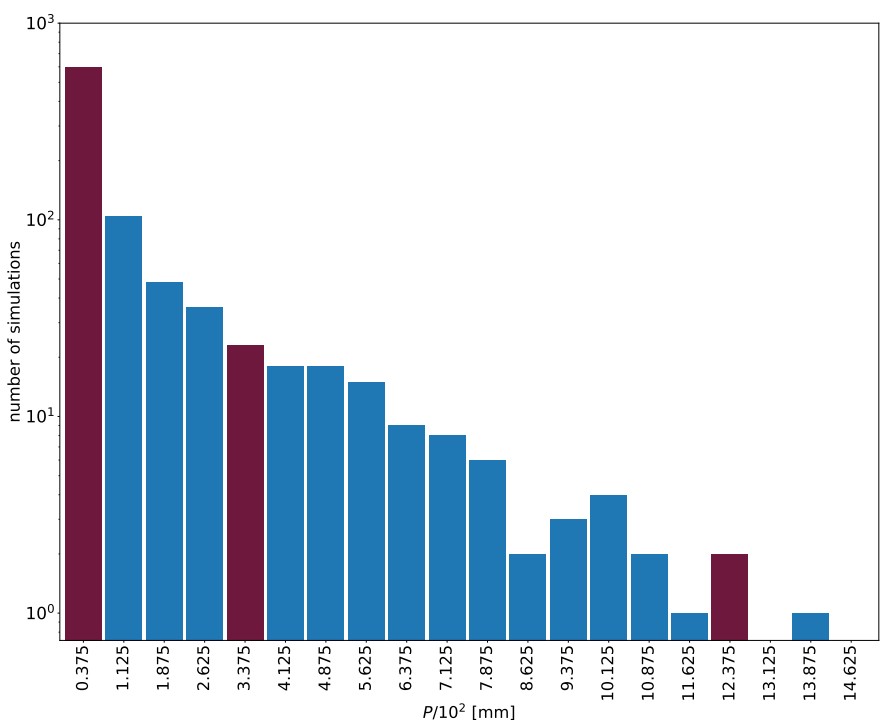

**Figure 6.** Frequency histogram of accumulated surface precipitation ($P$) from the ensemble of dynamic simulations with $N_{\mathrm{SD}}^{(\mathrm{bin})} = 10^2$. Horizontal axis is bin center. Bin width is $7.5 \cdot 10^{-3}$mm. The vertical axis is the number of simulations with $P$ within a bin. The bins in burgundy show the expected rain amount for LR, MR and HR velocity fields (left to right).

results are ensemble averages. For brevity, only results for $N_{\mathrm{SD}}^{(\mathrm{bin})} = 100$ are shown. Time series for other values of $N_{\mathrm{SD}}^{(\mathrm{bin})}$ are
similar and are available as supplemental information.

Time series of CTH, cc and CWP are smoother in dynamic than in kinematic simulations. In dynamic simulations, there are differences between simulation runs at the moment when the cloud starts to develop. When averaged over simulation runs, the results are smooth. In kinematic simulations, cloud develops in a very similar way in all simulations within an ensemble. Therefore, the ensemble average resembles a single dynamic simulation in that it changes significantly at short time scales.
This illustrates that, unsurprisingly, CTH, cc and CWP are more sensitive to the air flow than to the realization of collision-coalescence.

In all scenarios, cloud starts to develop at around $1500\,\mathrm{s}$. Afterward, it deepens with time, reaching maximum cc at around $5500\,\mathrm{s}$ and maximum CWP between $7000\,\mathrm{s}$ and $8000\,\mathrm{s}$, depending on the case. In kinematic scenarios, rain appears shortly after CWP reaches its maximum. The cloud almost entirely disappears at around $9500\,\mathrm{s}$ (CWP close to 0). A second cloud starts to
develop near the end of the simulation, indicated by an increase in CWP. The differences in rain between LR, MR and HR are explained by differences in CWP, with highest CWP giving most rain. In MR and in HR, CWP steadily increases until rain is

**Table 1.** Configuration of CC simulations. Columns, from left to right, are: Configuration name; Type of flow field (dynamic or kinematic); Collision-coalescence on/off flag; Method for SD initialization; Model of SGS advection and aerosol relaxation method (discussed in Section 5.7).

| name | flow field | coalescence | SD initialization | SGS SD motion | aerosol relaxation |
|---|---|---|---|---|---|
| D no coalescence | D | no | "constant SD"-init | no | no |
| LR no coalescence | LR | no | "constant SD"-init | no | no |
| MR no coalescence | MR | no | "constant SD"-init | no | no |
| HR no coalescence | HR | no | "constant SD"-init | no | no |
| D | D | yes | "constant SD"-init | no | no |
| LR | LR | yes | "constant SD"-init | no | no |
| MR | MR | yes | "constant SD"-init | no | no |
| HR | HR | yes | "constant SD"-init | no | no |
| D SGS SD motion | D | yes | "constant SD"-init | OU | below 700 m |
| LR SGS SD motion | LR | yes | "constant SD"-init | OU | below 700 m |
| MR SGS SD motion | MR | yes | "constant SD"-init | OU | below 700 m |
| HR SGS SD motion | HR | yes | "constant SD"-init | OU | below 700 m |
| HR f-i | HR | yes | "constant SD" fixed-init | no | no |
| HR $\xi$-i | HR | yes | $\xi_{\mathrm{const}}$-init | no | no |

formed. The difference is that CWP and CTH reach higher values in HR than in MR. In LR, there are multiple local maxima of CWP, each of them smaller than the maxima in MR and HR.

### 5.4 Numerical convergence in simulations without collision-coalescence

To reliably study convergence of the collision-coalescence algorithm, we first need to make sure that simulations without the collision-coalescence process have converged. The time step for condensation used in all simulations, 0.1 s, was found to give converged results (not shown). Here we focus on convergence with the number of SDs. Time series of basic cloud properties (cloud water content, cloud cover and cloud top height) agree for $N_{\mathrm{SD}}^{(\mathrm{bin})} \geq 50$ (they are plotted in the Supplement). Convergence of the DSD is analyzed by comparing profiles of concentration, mean radius and relative dispersion of radius of cloud droplets (Fig. 8). Out of the three parameters, relative dispersion is slowest to converge as it requires $N_{\mathrm{SD}}^{(\mathrm{bin})} \geq 1000$.

Note that the relative dispersion around 0.2 is within the lower part of the range of values observed in cumuliform (Lu et al., 2013) and stratiform (Miles et al., 2000; Pawlowska et al., 2006) clouds. This indicates that the DSD is realistic, an important point because Lagrangian models tend to generate narrow DSD. A too narrow DSD would results in a too low rate of collision-coalescence, and for slow collision-coalescence it is more difficult to reach convergence (Dziekan and Pawlowska, 2017). Therefore, studying convergence for an unrealistically narrow DSD could give too strict convergence requirements.

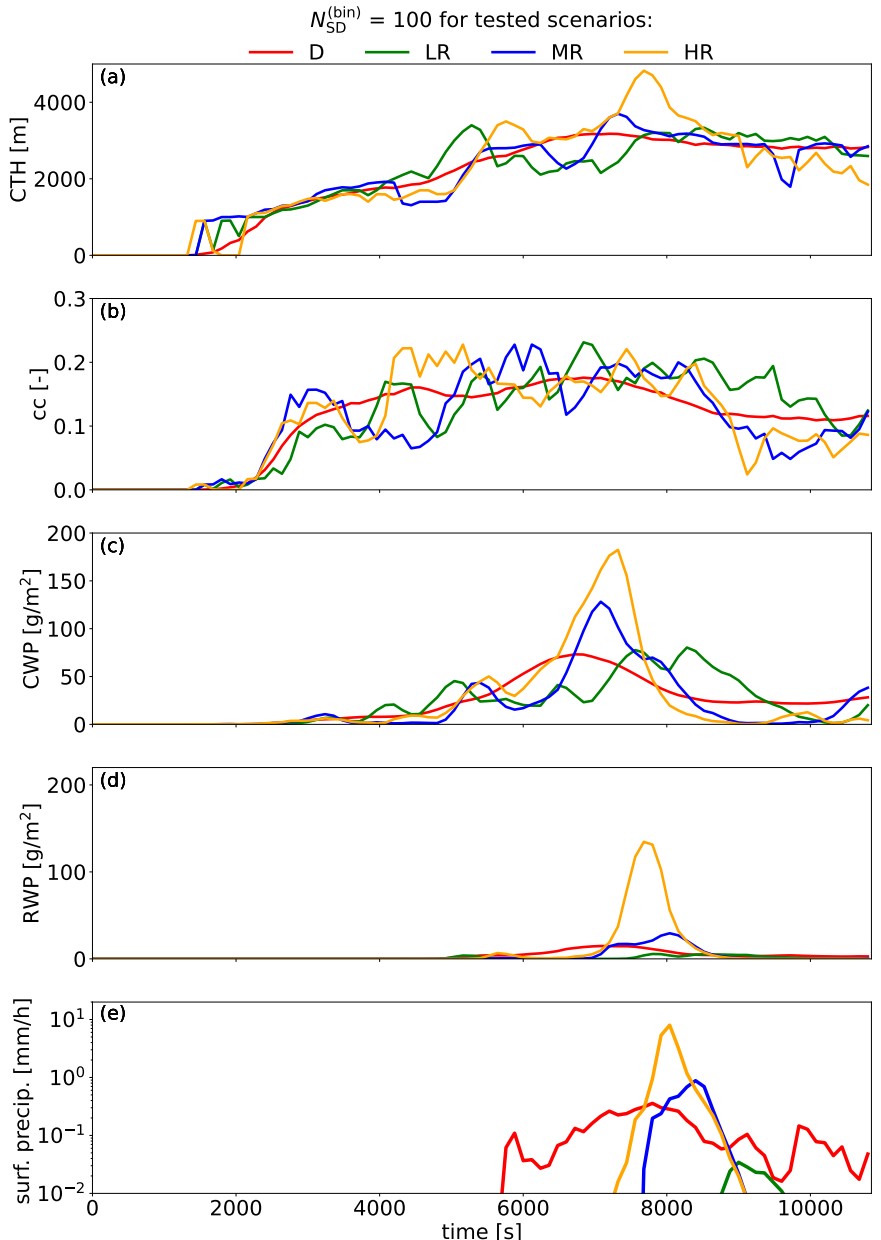

**Figure 7.** Time series of ensemble averages of cloud top height, cloud cover, cloud water path, rain water path and surface precipitation for D, LR, MR and HR scenarios with $N_{\mathrm{SD}}^{(\mathrm{bin})} = 100$. Cloud top height (CTH) is the vertical position of the topmost cloudy cell. Cloud cover (cc) is the fraction of columns with at least one cloudy cell. Cloudy cells are cells with cloud water mixing ratio greater than $10^{-5}$. Cloud droplets are droplets with $0.5\,\mu\mathrm{m} \leq r_w \leq 25\,\mu\mathrm{m}$. Rain drops are droplets with $25\,\mu\mathrm{m} \leq r_w$. Surface precipitation (surf. precip.), cloud water path (CWP) and rain water path (RWP) are domain averages divided by cc in order to obtain values representative of the cloudy area.

**Table 2.** Simulation ensemble sizes for all simulation types (defined in Table 1) and all tested values of $N_{SD}^{(bin)}$ and of $\Delta t_{coal}$.

| $N_{SD}^{(bin)}$ | $\Delta t_{coal}$ [s] | $\Omega_{LR}$ | $\Omega_{MR}$ | $\Omega_{HR}$ | $\Omega_D$ | no coalescence | | | | SGS SD motion | | | |
| | | | | | | $\Omega_{LR}$ | $\Omega_{MR}$ | $\Omega_{HR}$ | $\Omega_D$ | $\Omega_{HR}$ | $\Omega_D$ | $\Omega_{HR,f-i}$ | $\Omega_{HR,\xi-i}$ |
|---|---|---|---|---|---|---|---|---|---|---|---|---|---|
| $10^1$ | 0.1 | 200 | 200 | 201 | 600 | 1 | 1 | 1 | 100 | 151 | 251 | 100 | 201 |
| $5 \cdot 10^1$ | 0.1 | 150 | 201 | 201 | 601 | 1 | 1 | 1 | 101 | 0 | 0 | 0 | 0 |
| $10^2$ | 0.1 | 201 | 201 | 201 | 700 | 1 | 1 | 1 | 99 | 101 | 151 | 100 | 201 |
| $10^3$ | 0.1 | 50 | 50 | 31 | 301 | 1 | 1 | 1 | 101 | 51 | 101 | 100 | 101 |
| $5 \cdot 10^3$ | 0.1 | 0 | 0 | 100 | 0 | 0 | 0 | 0 | 0 | 0 | 0 | 37 | 50 |
| $10^4$ | 0.1 | 20 | 20 | 11 | 195 | 1 | 1 | 1 | 51 | 0 | 0 | 9 | 11 |
| $4 \cdot 10^4$ | 0.1 | 11 | 11 | 19 | 231 | 1 | 1 | 1 | 11 | 0 | 0 | 0 | 0 |
| $10^5$ | 0.1 | 6 | 6 | 11 | 109 | 0 | 0 | 0 | 0 | 0 | 0 | 0 | 0 |
| $10^2$ | 0.5 | 199 | 200 | 198 | 200 | 0 | 0 | 0 | 0 | 0 | 0 | 0 | 0 |
| $10^2$ | 0.05 | 199 | 200 | 201 | 200 | 0 | 0 | 0 | 0 | 0 | 0 | 0 | 0 |

## 5.5 Numerical convergence of precipitation

From now on, only simulations with collision-coalescence will be discussed. Profiles of precipitation flux for different number of SDs are shown in Fig. 9. We find that if there are differences between these profiles, they are similar at all levels (with very few exceptions). For example, at all levels $N_{SD}^{(bin)} = 100$ gives less precipitation than $N_{SD}^{(bin)} = 10^4$. This shows that

convergence of precipitation at one level is representative of convergence in the entire cloud. Therefore, we choose to study surface precipitation in detail, in particular the accumulated surface precipitation $P$. We denote ensemble mean with $\langle P \rangle$ and ensemble standard deviation with $\sigma(P)$. For estimating errors of ensemble statistics, we use the following formulas. The standard error of $\langle P \rangle$ is:

$$\text{se}(\langle P \rangle) = \frac{\sigma(P)}{\sqrt{n}}, \tag{3}$$

where $n$ is ensemble size. The standard error of $\sigma(P)$ is (Rao, 1973, p.438):

$$\text{se}(\sigma(P)) = \frac{1}{2\sigma(P)} \sqrt{\frac{1}{n}\left(\left\langle (P - \langle P \rangle)^4 \right\rangle - \frac{n-3}{n-1}\sigma(P)^4\right)}. \tag{4}$$

The 95 % confidence interval of $\langle P \rangle$ is:

$$\text{CI}_{95\%}(\langle P \rangle) = [\langle P \rangle - 1.96 \cdot \text{se}(\langle P \rangle), \langle P \rangle + 1.96 \cdot \text{se}(\langle P \rangle)]. \tag{5}$$

The 95 % confidence interval of $\sigma(P)$ is (Sheskin, 2020, p.217):

$$\text{CI}_{95\%}(\sigma(P)) = \left[\sigma(P)\sqrt{\frac{n-1}{f(0.975, n-1)}}, \sigma(P)\sqrt{\frac{n-1}{f(0.025, n-1)}}\right], \tag{6}$$

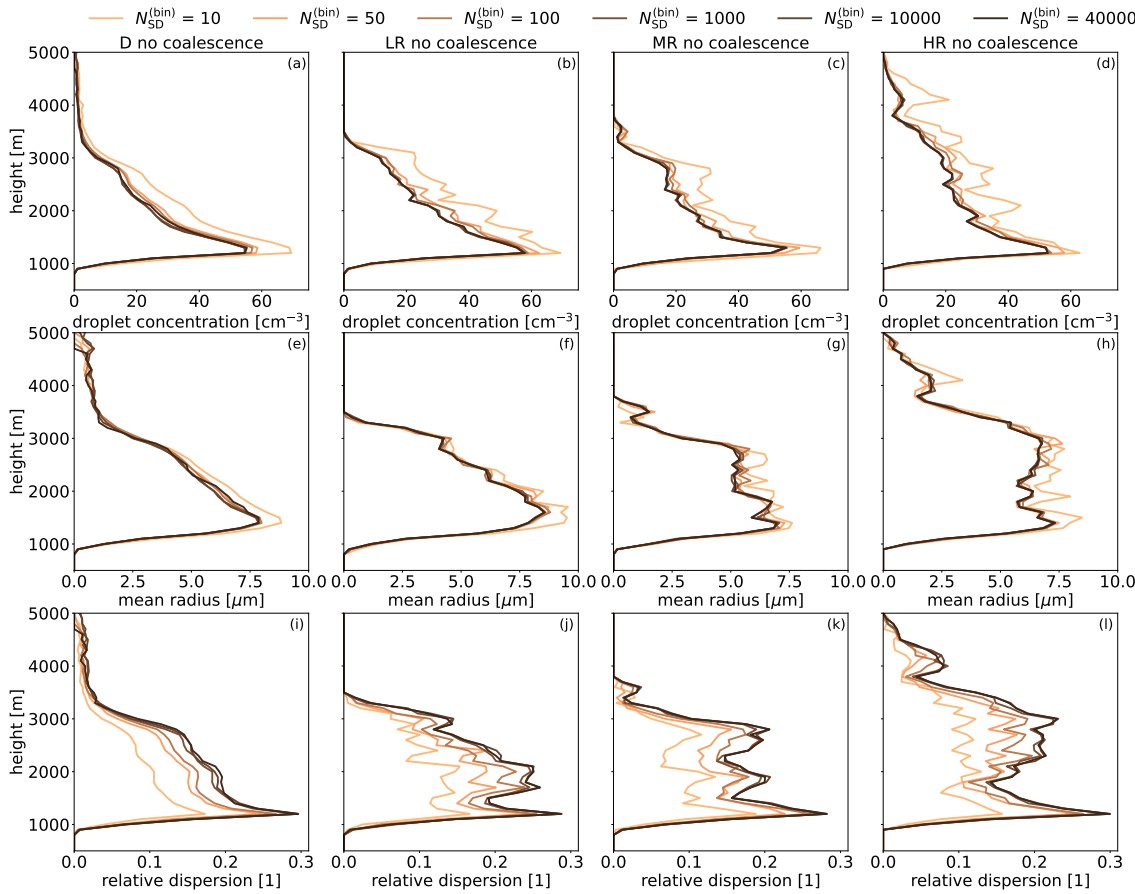

**Figure 8.** Profiles of cloud droplet concentration (top row), cloud droplet mean radius (center row) and relative dispersion of cloud droplet radius (bottom row) from simulations without collision-coalescence. Profiles are averaged over cloudy cells, over the time interval between $1800\,\mathrm{s}$ and $9600\,\mathrm{s}$, and over the ensemble of simulations.

where $f(x, y)$ is the inverse CDF of the chi-squared distribution.

Figure 10 shows sensitivity of surface precipitation to $\Delta t_{\mathrm{coal}}$, the time step with which coalescence is modeled. We find no statistically significant impact of $\Delta t_{\mathrm{coal}}$ on $\langle P \rangle$ or on $\sigma(P)$ for $\Delta t_{\mathrm{coal}} \leq 0.5\,\mathrm{s}$. Sensitivity to time step was tested only for $N_{\mathrm{SD}}^{(\mathrm{bin})} = 100$, because box simulations showed that results converge for the same value of $\Delta t_{\mathrm{coal}}$, independent of the value of $N_{\mathrm{SD}}^{(\mathrm{bin})}$. Study of sensitivity to $N_{\mathrm{SD}}^{(\mathrm{bin})}$ that is discussed next was done for $\Delta_{\mathrm{coal}} = 0.1\,\mathrm{s}$.

Mean surface precipitation for differing number of SDs is shown in Fig. 11. We find that $\langle P \rangle$ varies with $N_{\mathrm{SD}}^{(\mathrm{bin})}$ in a non-trivial way, similar in all four scenarios. Mean precipitation is the highest for $N_{\mathrm{SD}}^{(\mathrm{bin})} = 10$. Then, there is a large decrease in $\langle P \rangle$ when $N_{\mathrm{SD}}^{(\mathrm{bin})}$ is increased from 10 to 50. A minimum of $\langle P \rangle$ is found between $N_{\mathrm{SD}}^{(\mathrm{bin})} = 50$ and $N_{\mathrm{SD}}^{(\mathrm{bin})} = 10^3$, depending on the scenario. Beyond this minimum, $\langle P \rangle$ slowly increases (see subplots e-h). In the D and HR scenarios, there is evidence for convergence of $\langle P \rangle$ for $N_{\mathrm{SD}}^{(\mathrm{bin})} \geq 10^4$ and for $N_{\mathrm{SD}}^{(\mathrm{bin})} \geq 5 \times 10^3$, respectively. Above these values, centers of confidence

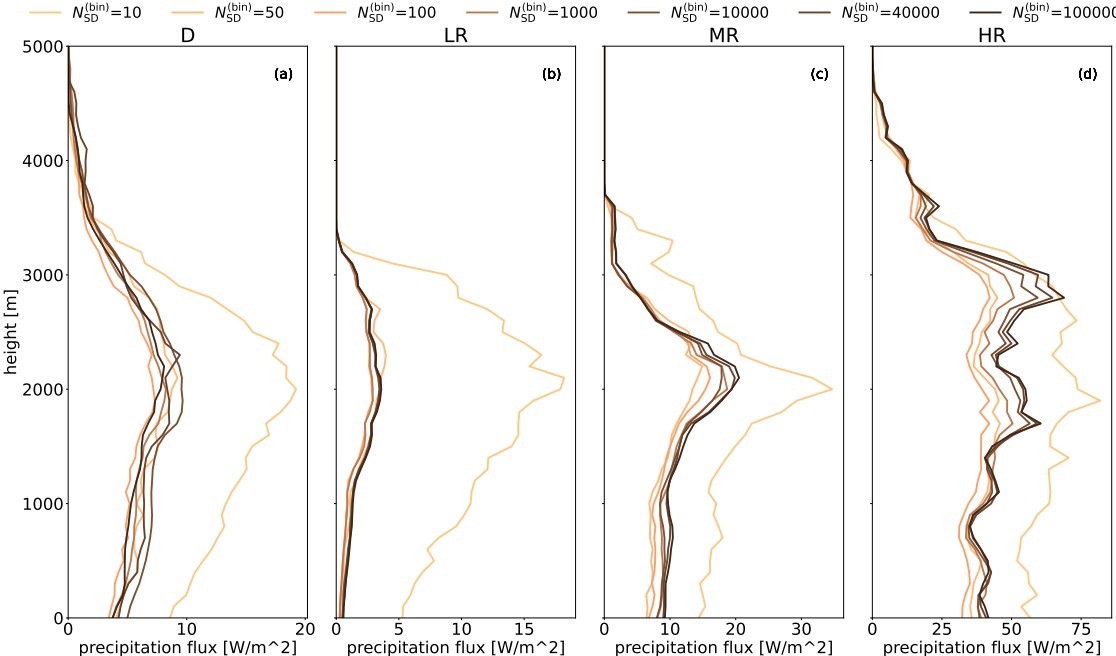

**Figure 9.** Profiles of precipitation flux in simulations with collision-coalescence. Profiles are averaged over all cells, over the time interval between $1800\,\mathrm{s}$ and $9600\,\mathrm{s}$, and over the ensemble of simulations. Time step for coalescence is $\Delta_{\mathrm{coal}} = 0.1\,\mathrm{s}$.

intervals are at similar positions and the intervals overlap. In the LR and MR scenarios, the center of the confidence interval increases for $N_{\mathrm{SD}}^{(\mathrm{bin})} \geq 10^3$. Although the intervals often overlap, this systematic increase indicates that $\langle P \rangle$ has not converged.

Changes of $\langle P \rangle$ for $N_{\mathrm{SD}}^{(\mathrm{bin})} \leq 10^3$ are consistent with changes of the mean DSD in box simulations (see Fig. 3 (d)). In box simulations, $N_{\mathrm{SD}}^{(\mathrm{bin})} = 10$ gives too few droplets with radii between 40 and 120 microns, but too many droplets with radii greater than $120\,\mu\mathrm{m}$. Since surface precipitation is sensitive to the largest droplets, this is consistent with too large $\langle P \rangle$ seen in CC simulations. For $N_{\mathrm{SD}}^{(\mathrm{bin})} = 10^2$, number of the largest droplets is no longer overestimated in box simulations, but there are still too few droplets with radii between 50 and 120 microns. This is consistent with a sharp decrease of $\langle P \rangle$ between $N_{\mathrm{SD}}^{(\mathrm{bin})} = 10$ and $N_{\mathrm{SD}}^{(\mathrm{bin})} = 10^2$. In box simulations with $N_{\mathrm{SD}}^{(\mathrm{bin})} = 10^3$ the number of droplets with sizes between 50 and 120 microns is no longer underestimated, what is consistent with an increase of $\langle P \rangle$ between $N_{\mathrm{SD}}^{(\mathrm{bin})} = 10^2$ and $N_{\mathrm{SD}}^{(\mathrm{bin})} = 10^3$ in CC simulations.

The increase of $\langle P \rangle$ for $N_{\mathrm{SD}}^{(\mathrm{bin})} > 10^3$ cannot however be easily explained by box simulations nor by CC simulations without collision-coalescence, because mean results of these two simulation types converge for $N_{\mathrm{SD}}^{(\mathrm{bin})} = 10^3$ (see Section 3 and Section 5.4). This suggests that $\langle P \rangle$ may be affected by too large variance of the DSD, which does not converge in box simulations even for $N_{\mathrm{SD}}^{(\mathrm{bin})} > 10^3$. The fact that $\langle P \rangle$ quickly converges with $\Delta t_{\mathrm{coal}}$ is consistent with this hypothesis, because $\Delta t_{\mathrm{coal}}$ does not affect the variance of the DSD. Too large variance and correct mean of the DSD in box simulations corresponds to a situation in which in LES differences between DSD in neighboring cells are larger than expected. There are some cells with more large droplets than expected, and some cells with fewer large droplets than expected. To show how the spatial distribution

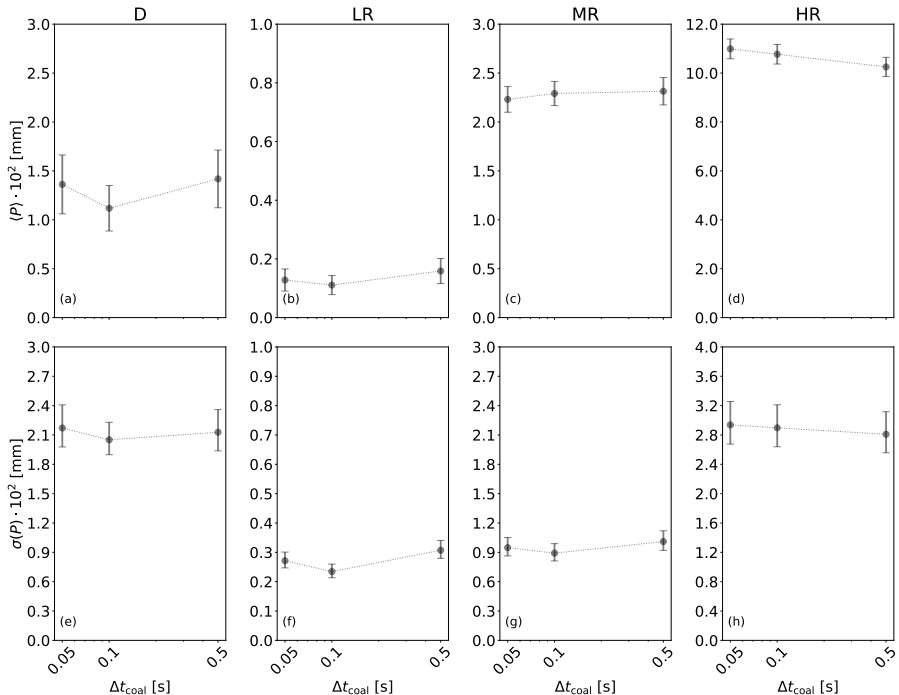

**Figure 10.** Ensemble mean and standard deviation of accumulated precipitation at the end of a simulation against time step for coalescence in four scenarios of a CC simulation with $N_{\mathrm{SD}}^{(\mathrm{bin})} = 100$. Error bars represent the 95% confidence interval.

of rain within the cloud changes with $N_{\mathrm{SD}}^{(\mathrm{bin})}$, in Fig. 12 we show the probability density function of rain water content at four points in time, from just before the onset of surface precipitation until its maximum. We find that the distribution narrows with increasing $N_{\mathrm{SD}}^{(\mathrm{bin})}$. For small $N_{\mathrm{SD}}^{(\mathrm{bin})}$, the distribution is bimodal, in particular at earlier times. As $N_{\mathrm{SD}}^{(\mathrm{bin})}$ is increased,
the smaller mode disappears, and we get a single mode, with a maximum for smaller values of rain water content than the maximum of the larger of the two modes observed for small $N_{\mathrm{SD}}^{(\mathrm{bin})}$. The distribution converges for a similar value of $N_{\mathrm{SD}}^{(\mathrm{bin})}$ as the one required for convergence of $\langle P \rangle$.

The standard deviation of precipitation for differing number of SDs is shown in Fig. 13. In dynamic simulations (subplot a), $\sigma(P)$ is large for $N_{\mathrm{SD}}^{(\mathrm{bin})} = 10$, then sharply decreases for $N_{\mathrm{SD}}^{(\mathrm{bin})} = 50$ and does not change significantly as $N_{\mathrm{SD}}^{(\mathrm{bin})}$ is further
increased. Most of the 95 % confidence intervals are overlapping for $N_{\mathrm{SD}}^{(\mathrm{bin})} \geq 50$. The relative standard deviation (subplot e) is around 1.5 for $N_{\mathrm{SD}}^{(\mathrm{bin})} \geq 50$. The relatively low sensitivity of $\sigma(P)$ to $N_{\mathrm{SD}}^{(\mathrm{bin})}$ in dynamic simulations shows that precipitation is more sensitive to differences in the flow field, which can be a consequence of small random perturbations of initial conditions, than to differences in realization of the collision-coalescence model of particle microphysics.

In kinematic simulations (subplots b-d) standard deviation of precipitation is more sensitive to $N_{\mathrm{SD}}^{(\mathrm{bin})}$ than in dynamic
simulations. There is a significant decrease of $\sigma(P)$ as $N_{\mathrm{SD}}^{(\mathrm{bin})}$ is increased (except for small $N_{\mathrm{SD}}^{(\mathrm{bin})}$ in HR). The relative standard deviation has a maximum for $N_{\mathrm{SD}}^{(\mathrm{bin})}$ between 50 and 100, and decreases for higher $N_{\mathrm{SD}}^{(\mathrm{bin})}$ (subplots f-h). This shows

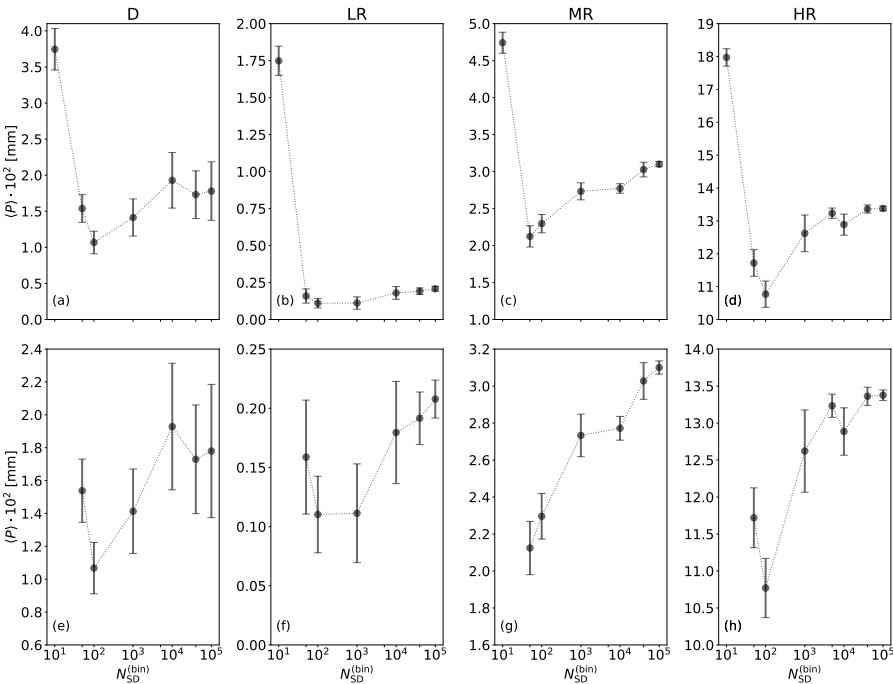

**Figure 11.** Ensemble mean of precipitation against number of super-droplets for four scenarios: D, LR, MR and HR. In (e-h) the same results are shown as in (a-d), but without $N_{\mathrm{SD}}^{(\mathrm{bin})} = 10$. Error bars show the $95\%$ confidence interval.

that in the absence of differences in flow field, precipitation is governed by realizations of the collision-coalescence model. Comparing D with MR, which is the kinematic case with the most similar $\langle P \rangle$, we find that $\sigma(P)/\langle P \rangle$ in dynamic simulations is much larger than in kinematic simulations, from around 4 times larger for low $N_{\mathrm{SD}}^{(\mathrm{bin})}$ to up to 70 times larger for large $N_{\mathrm{SD}}^{(\mathrm{bin})}$. This supports the conclusion that precipitation primarily depends on the realized flow field and not on the realization of AON.

## 5.6 Sensitivity to SD initialization method

Collision-coalescence in particle microphysics is sensitive to the way SD attributes are initialized. Therefore, the way precipitation changes with the number of SDs could depend on SD initialization. To check this, we test convergence for three types of SD initialization that were introduced in Section 2.1: $\xi_{\mathrm{const}}$-init, "const SD"-init and "const SD" fixed-init. In "const SD" fixed-init the outermost bin edges for dry radius were set to $1\,\mathrm{nm}$ and $5\,\mu\mathrm{m}$. For all methods, the initial DSD averaged over a large number of cells agrees very well with the prescribed distribution (Fig. 2 (a) shows this, albeit for a different distribution). All methods give very good representation of the initial DSD. Comparison of results for different initialization methods in the HR case is shown in Fig. 14. We see only minor differences between "const SD"-init and "const SD" fixed-init. Both methods use bins to make sampling of the initial aerosol radius more even, but differ in the way the entire bin range is selected. Re-

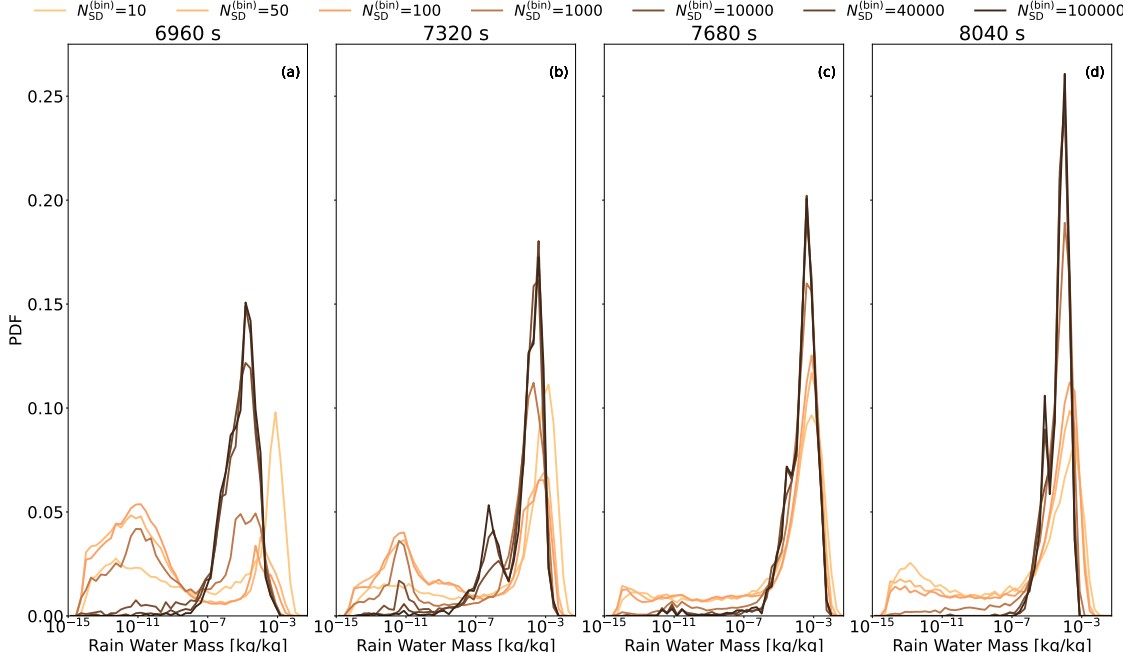

**Figure 12.** Probability density function of rain water mass in cloudy cells at four moments in time, averaged from the HR simulation ensemble.

cently, Hill et al. (2023) found differences in precipitation between different implementations of particle microphysics, both using AON and binned initialization. Differences in details of bin initialization were proposed as one of potential reasons for the observed discrepancies. Good agreement between "const SD"-init and "const SD" fixed-init in our simulations suggests that some other factor is responsible for the discrepancies discussed in Hill et al. (2023).

The $\xi_{\mathrm{const}}$-init gives significantly different results than the bin initialization methods. In $\xi_{\mathrm{const}}$-init there is very little precipitation when $N_{\mathrm{SD}}^{(\mathrm{init})}$ is small. As more SDs are used, the amount of precipitation increases. It is plausible that all methods of initialization should converge for large enough number of SDs. However, even for $N_{\mathrm{SD}}^{(\mathrm{init})} = 10^4$, which was the largest number of SDs that we were able to model in $\xi_{\mathrm{const}}$-init, $\xi_{\mathrm{const}}$-init gives less precipitation than the other methods. Unterstrasser et al. (2017) showed that $\xi_{\mathrm{const}}$-init requires a huge number of SDs in box simulations of collision-coalescence, and the authors

hypothesized that it may require fewer SDs in cloud simulations. Our results show that this is not the case: in 2D simulations $\xi_{\mathrm{const}}$-init has the same deficiencies as in box simulations. It requires a very large number of super-droplets, unattainable in 3D LES, to get convergence in precipitation. For fewer SDs, it gives significantly too little precipitation.

### 5.7   Sensitivity to SGS motion of SDs

In multi-box simulations, mixing of droplets between cells helps achieve convergence of collision-coalescence modeling. In

CC simulations discussed so far, intercell mixing was caused by the resolved-scale motion and by sedimentation, but there was

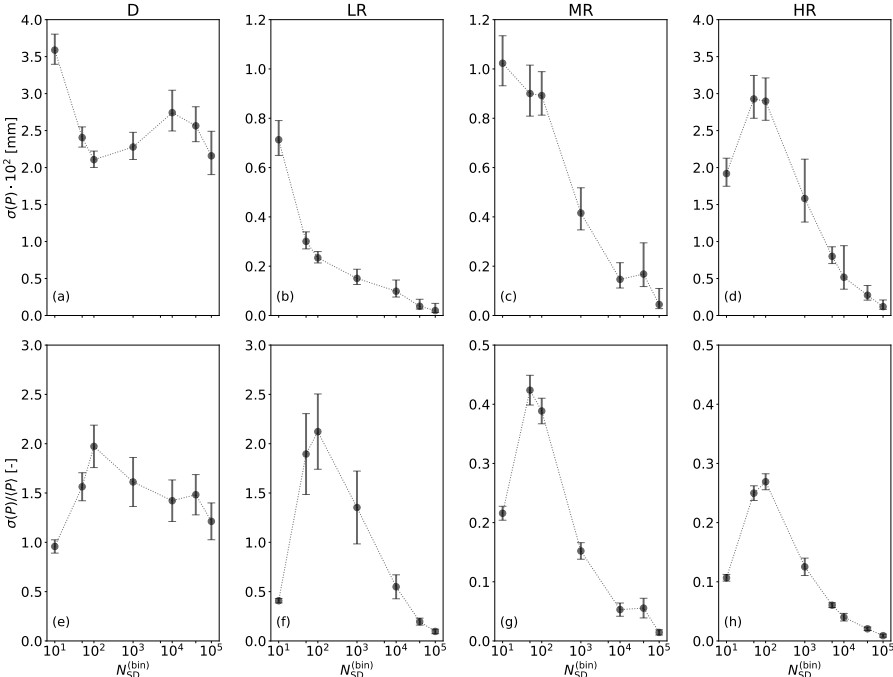

**Figure 13.** Ensemble standard deviation (a-d) and relative standard deviation(e-h) of precipitation against number of super-droplets for four types of simulations: D, LR, MR and HR. In (a-d), error bars show the $95\%$ confidence interval. In (e-h), error bars show the error $\mathrm{e}\left(\sigma(P)/\langle P\rangle\right)$ estimated with: $\frac{\mathrm{e}(\sigma(P)/\langle P\rangle)}{\sigma(P)/\langle P\rangle} = \sqrt{\left(\frac{\mathrm{se}(\langle P\rangle)}{\langle P\rangle}\right)^2 + \left(\frac{\mathrm{se}(\sigma(P))}{\sigma(P)}\right)^2}$.

no SGS motion of SDs. Here, we consider simulations in which SGS velocity of SDs is modeled using the OU method, what should make mixing more efficient. In the OU model, we assume a constant and uniform TKE dissipation rate of $5\,\mathrm{cm^2s^{-3}}$. Subgrid-scale motion can increase the number of SDs that hit the bottom wall of the domain, resulting in a decrease in the number of SDs and in aerosol concentration. This complicates the comparison with simulations without SGS motion of SDs.

Therefore, in simulations with the OU model we compensate for aerosol depletion by adding SDs whenever there are fewer aerosols than at simulation start. Details of the procedure can be found in Dziekan et al. (2021). This relaxation is done only in the lowest $700\,\mathrm{m}$ of the domain, below the cloud base. Parameters of relaxation were tuned to get SD number and aerosol concentration close to those in simulations without SGS motion of SDs. Histograms with the number of SDs in cloudy cells and vertical profiles of several parameters, including aerosol concentration, are shown in the Supplement. Convergence of

precipitation statistics with and without the OU model is compared in Fig. 15. We don't see any significant effect of SGS motion. In HR, mean precipitation agrees very well. In D, there are small differences, but $\langle P\rangle$ changes with $N_{\mathrm{SD}}^{(\mathrm{bin})}$ similarly with and without the OU model. As we didn't observe any impact of SGS SD motion for $N_{\mathrm{SD}}^{(\mathrm{bin})} \leq 10^4$, we did not test higher $N_{\mathrm{SD}}^{(\mathrm{bin})}$.

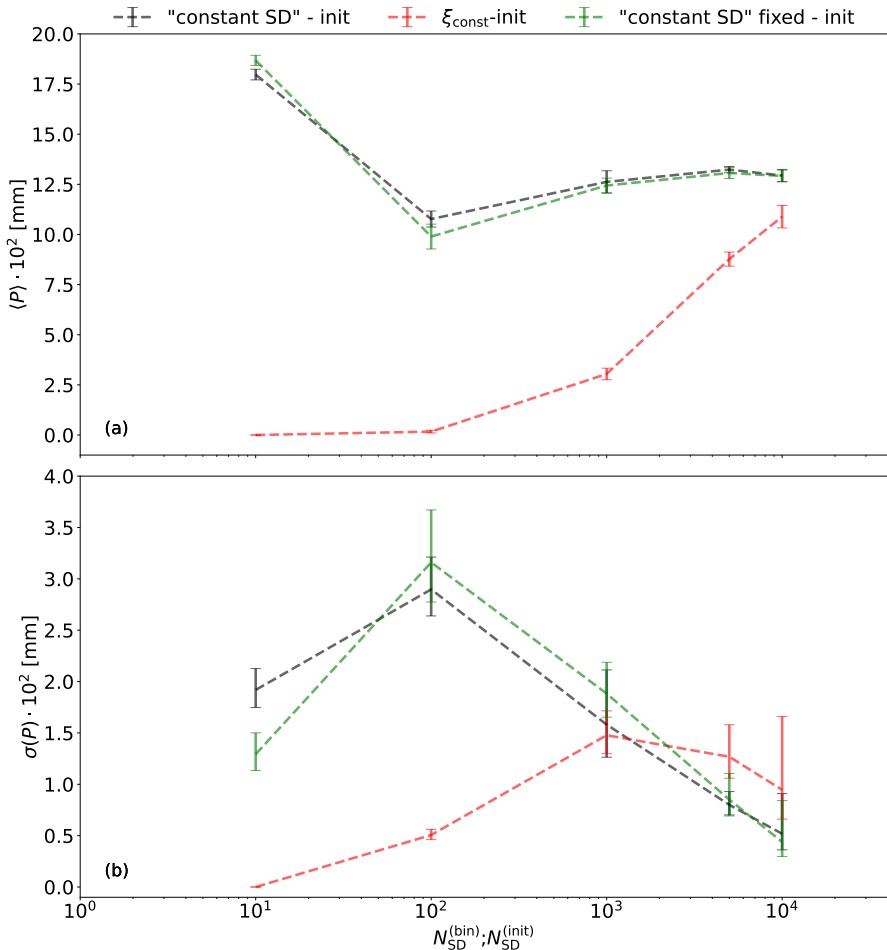

**Figure 14.** Mean $\langle P \rangle$ and standard deviation $\sigma(P)$ of accumulated precipitation against the number of SDs for HR simulations with three SD initialization methods. Horizontal axis is $N_{\mathrm{SD}}^{(\mathrm{bin})}$ in "constant SD"-init and $N_{\mathrm{SD}}^{(\mathrm{init})}$ in $\xi_{\mathrm{const}}$-init. In "constant SD" fixed-init we have $N_{\mathrm{SD}}^{(\mathrm{init})} = N_{\mathrm{SD}}^{(\mathrm{bin})}$. Error bars represent the $95\%$ confidence interval.

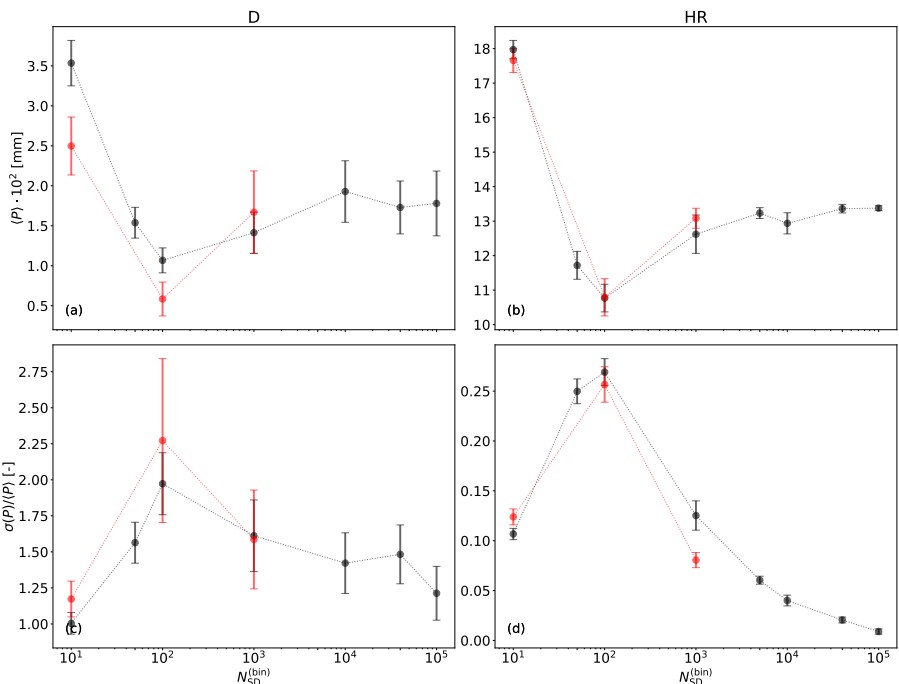

**Figure 15.** Mean and relative standard deviation of accumulated precipitation against the number of SDs for simulations with (red) and without (black) the OU model of SGS motion of SDs.

## 6 Conclusions

Our study shows that using particle microphysics it is more difficult to reach numerical convergence of precipitation in cloud simulations than it is to reach convergence of mean DSD in an ensemble of box or multi-box simulations of collision-coalescence. In general, convergence requirements are less strict in strongly precipitating clouds than in lightly precipitating clouds.

It is relatively easy to have convergence with $\Delta t_{\text{coal}}$. Mean precipitation in our isolated cumulus simulations converged for $\Delta t_{\text{coal}} = 0.5\,\text{s}$. The same time step length was also sufficient in simulations of cumulus cloud fields (Dziekan et al., 2021). However, box simulations presented in this text and stratocumulus cloud field simulations from Dziekan et al. (2021) required $\Delta t_{\text{coal}} = 0.1\,\text{s}$. This suggests that $\Delta t_{\text{coal}} = 0.1\,\text{s}$ is a safe choice for cloud modeling. We used the linear sampling technique. Quadratic sampling may allow for longer time steps (Unterstrasser et al., 2020). Variance of precipitation in cloud simulations and variance of the DSD in box simulation are not sensitive to $\Delta t_{\text{coal}}$.

It is more difficult to reach convergence with the number of SDs per cell. In box simulations, mean DSD converges for $N_{\text{SD}}^{(\text{bin})} \geq 10^3$, but variance of the DSD decreases with $N_{\text{SD}}^{(\text{bin})}$ without converging. In multi-box simulations (multiple boxes with mixing between them) mean DSD converges for fewer SDs than in box simulations, what is attributed to a positive role of intercell mixing. In cumulus simulations, mean precipitation converges for $N_{\text{SD}}^{(\text{bin})} \geq 5 \times 10^3$ in the most heavily precipi-

tating kinematic case and for $N_{\text{SD}}^{(\text{bin})} \geq 10^4$ in dynamic simulations. In kinematic cases with less precipitation, we do not see

convergence of mean precipitation even for $N_{\text{SD}}^{(\text{bin})} = 10^5$.

     It is not clear why convergence is slower in CC than in box simulations. In CC, convergence of mean precipitation coincides with convergence of the spatial distribution of rain. This may suggest that mean precipitation is dependent on spatial distribution of droplet sizes, probably because of interaction between cells. However, in multi-box simulations we observe that intercell mixing helps reach convergence. Increasing the rate of intercell mixing in CC by using a SGS model does not help with

convergence. This does not necessarily indicate that intercell mixing is not important for convergence in CC. It is possible that the increase in intercell mixing caused by the SGS model is small in relation to interecell mixing caused by resolved eddies and by sedimentation. Precipitation is sensitive to the super-droplet initialization procedure. In this study initial radii were almost evenly distributed on a logarithmic scale. If droplet radii are randomly drawn from the initial distribution, it is more difficult to reach convergence with $N_{\text{SD}}^{(\text{bin})}$ and using too small $N_{\text{SD}}^{(\text{bin})}$ induces larger errors. Note that the splitting-merging algorithm

of Schwenkel et al. (2018), which was not included in this study, could help achieve convergence.

     Variance of precipitation in an ensemble of cloud simulations decreases with $N_{\text{SD}}^{(\text{bin})}$, but only if the same flow field is used in the ensemble. If the flow field is different in different simulations, e.g. due to random perturbations of initial conditions, variance of precipitation is not sensitive to $N_{\text{SD}}^{(\text{bin})}$. This shows that in typical LES, the increased variance in the number of collisions in particle microphysics does not affect variability in rain between simulation runs, because differences in realized

flow fields are more important.

## Appendix A: Periodic synthetic turbulence model

In multi-box simulations, incompressible isotropic turbulence is modeled as a sum of random Fourier modes. The model is similar to that used in Sidin et al. (2009), but the generated velocity field is periodic because of the boundary conditions in multi-box simulations. We shall assume periodic boundary conditions in the space variable $\boldsymbol{r} = (x, y, z)$,

$$\boldsymbol{u}(x + n_x L, y + n_y L, z + n_z L) \equiv \boldsymbol{u}(\boldsymbol{r} + \boldsymbol{n}L) = \boldsymbol{u}(x, y, z), \tag{A1}$$

for all $x$, $y$, $z$ and all signed integer $n_x, n_y, n_z$, where $L$ is called the period. It is enough to consider the restriction of the flow into a periodic cubic box of side $L$. Using Fourier series we write the velocity field as

$$\boldsymbol{u}(\boldsymbol{r}, t) = \sum_{\boldsymbol{k}} \hat{\boldsymbol{u}}(\boldsymbol{k}, t) \exp[i\boldsymbol{k} \cdot \boldsymbol{r}] \tag{A2}$$

with

$$\boldsymbol{k} = \frac{2\pi}{L}\boldsymbol{n} = \frac{2\pi}{L}(n_x, n_y, n_z). \tag{A3}$$

Using the incompressibility condition, and since $\boldsymbol{u}(\boldsymbol{r}, t)$ is real-valued, the velocity field (A2) reduces to:

$$\boldsymbol{u}(\boldsymbol{r}, t) = \sum_{n=1}^{N} \sum_{m=-M(n)/2}^{M(n)/2} [\boldsymbol{a}_{n,m}(t) \times \hat{\boldsymbol{k}}_{n,m}] \cos(\boldsymbol{k}_{n,m} \cdot \boldsymbol{r}) - [\boldsymbol{b}_{n,m}(t) \times \hat{\boldsymbol{k}}_{n,m}] \sin(\boldsymbol{k}_{n,m} \cdot \boldsymbol{r}), \tag{A4}$$

where $\hat{k} = k/|k|$ is the unit vector in the direction of $k$ and $M(n)$ is the $n$-dependent *multiplicity* (or *degeneracy*) of wavevectors $k_{n,m}$ with the same magnitude $k_n = |k_{n,m}|$. For a given value of $n$ (numbering the magnitude $k_n = |k_{n,m}|$), the index $m$ numbers the wavevectors $k_{n,m}$ with

$$k_{n,-m} = -k_{n,m}, \quad m = 1,\ldots,\tfrac{1}{2}M(n). \tag{A5}$$

The time evolution of the random vector coefficients $a_{n,m}(t)$ and $b_{n,m}(t)$ reads

$$a_{n,m}(t+\delta t) = r_n a_{n,m}(t) + \sigma_n \sqrt{1-r_n^2}\, \xi_a, \tag{A6}$$

$$b_{n,m}(t+\delta t) = r_n b_{n,m}(t) + \sigma_n \sqrt{1-r_n^2}\, \xi_b, \tag{A7}$$

for $m = 1,\ldots,M(n)/2$. In the expression above, $\xi_a$ and $\xi_b$ are independent random vectors, with components taken from a Gaussian distribution with zero-mean and unit variance. The values of $a_{n,m}(t)$ and $b_{n,m}(t)$ for $m < 0$ are obtained from:

$$a_{n,-m}(t) = -a_{n,m}(t), \quad b_{n,-m}(t) = b_{n,m}(t). \tag{A8}$$

The remaining quantities are: the relaxation function

$$r_n = \exp(-\omega_n \delta t), \tag{A9}$$

with frequencies

$$\omega_n \sim \sqrt{k_n^3 E(k_n)}, \tag{A10}$$

Kolmogorov energy spectrum in the inertial subrange

$$E(k_n) \sim k_n^{-5/3}, \tag{A11}$$

variances

$$\sigma_n^2 = E(k_n)\Delta k_n/M(n) \tag{A12}$$

and differences in wavevector magnitudes

$$\Delta k_n = \frac{1}{2}(k_{n+1} - k_{n-1}), \quad 2 \le n \le N-1, \tag{A13}$$

$$\Delta k_1 = \frac{1}{2}(k_2 - k_1), \quad \Delta k_N = \frac{1}{2}(k_N - k_{N-1}). \tag{A14}$$

Validity of the periodic model is tested by comparing pair separation statistics with results from the non-periodic model of Sidin et al. (2009). Initial pair separation is equal to the Kolmogorov length assumed to be $\eta = 1\,\mathrm{mm}$. The size of the largest eddies is $L = 1\mathrm{m}$. In the non-periodic model, 200 wavevector magnitudes were used that form a geometric series between $L$ and $\eta$ (Sidin et al., 2009). For each wavevector magnitude, 50 wavevectors were randomly selected. In the periodic model, we used magnitudes of all periodic wavevectors between $L$ and $\eta$. For each wavevector magnitude, $n$ we randomly selected $\min(10, M(n))$ wavevectors, where $M(n)$ is degeneracy. Time step was $0.1\mathrm{s}$. Results are plotted in Fig. A1. We find that pair separation is in the periodic model is somewhat larger. However, given that the choice of $\epsilon$ in multi-box tests is arbitrary, we decide that the periodic model is sufficiently realistic.

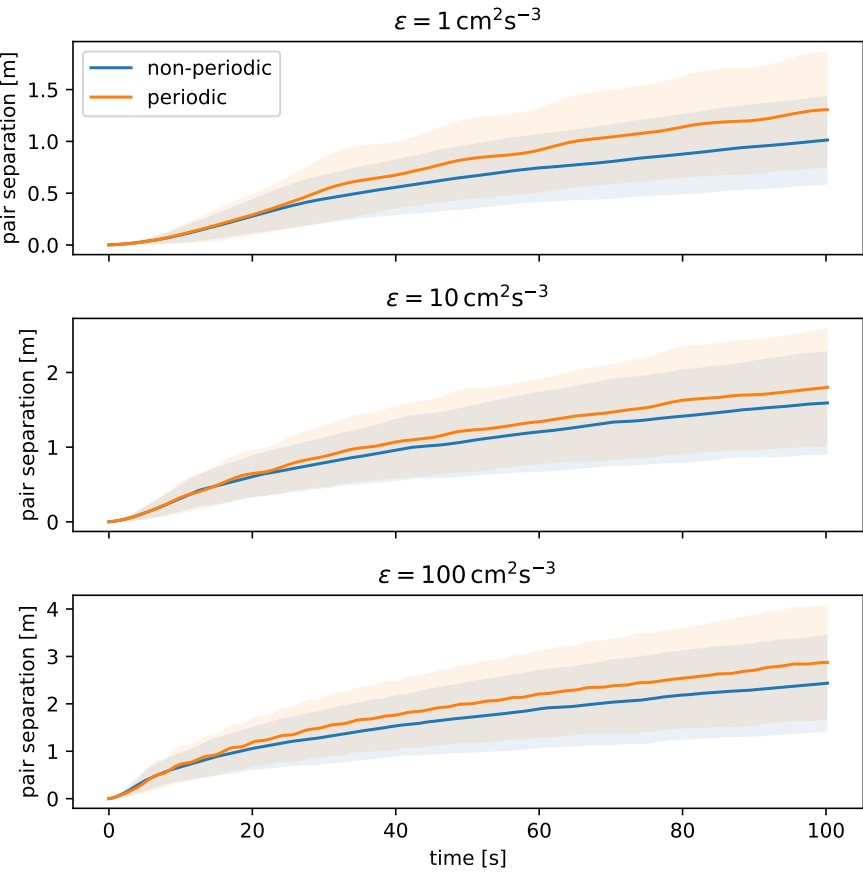

**Figure A1.** Pair separation statistics from the periodic and non-periodic synthetic turbulence models for different values of the TKE dissipation rate. Solid lines show ensemble mean, and shading shows one standard deviation interval.

*Author contributions.* PZ and PD conceived the idea of the study. Simulations, analyses and data visualisation were done by PZ for CC simulations and by PD for box and multi-box simulations. All authors contributed to the manuscript. Funding were secured by PD and HP.

*Competing interests.* No competing interests are present

*Code and data availability.* Cloud simulations were done using UWLCM, box and multi-box were done with *coal_fluctu*. These models use the SDM implementation from the *libcloudph++* library (Arabas et al., 2015). UWLCM also uses the *libmpdata++* library (Jaruga et al., 2015). Plotting of UWLCM results was done with the *UWLCM_plotting* package. Pair separation tests were done using *synth_turb*. In the study, the following code versions were used: UWLCM v2.1 (Dziekan et al., 2023), libmpdata++ v2.1 (Arabas et al., 2023b),

libcloudph++ v3.1 (Arabas et al., 2023a), UWLCM_plotting v1.0 (Dziekan and Zmijewski, 2023), coal_fluctu v2.2 (Dziekan, 2023a),
synth_turb v0.1 (Dziekan, 2023b). Dataset, run scripts, and plotting scripts are available at Zmijewski et al. (2023).

*Acknowledgements.*   We thank Dr Gustavo Abade for help with the periodic synthetic turbulence model and its description. This research was
supported by the Polish National Science Center grant no 2018/31/D/ST10/01577 and no 2016/23/B/ST10/00690. We gratefully acknowledge
Poland's high-performance Infrastructure PLGrid (HPC Centers: ACK Cyfronet AGH, PCSS, CI TASK, WCSS) for providing computer
facilities and support within computational grant no. PLG/2022/015886. The calculations were made with the support of the Interdisciplinary
Center for Mathematical and Computational Modeling of the University of Warsaw (ICM UW) under the computational grant no GR84-48.

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
