# Peer review of "Modeling Collision-Coalescence in Particle Microphysics: Numerical Convergence of Mean and Variance of Precipitation in Cloud Simulations Using University of Warsaw Lagrangian Cloud Model (UWLCM) 2.1"

_Geoscientific Model Development, 2023_

## Referee Comment (RC1)

**Review of "Modeling Collision-Coalescence in Particle Microphysics: Numerical Convergence of Mean and Variance of Precipitation in Cloud Simulations Using University of Warsaw Lagrangian Cloud Model (UWLCM) 2.1" by Zmijewski et al. (gmd-2023-44)**

The presented study investigates the numerical convergence of cumulus congestus simulations with a Lagrangian (or particle-based) cloud microphysics scheme. The authors find that the convergence requires much more computational particles than previously assumed. In fact, the authors suggest that convergence is only possible using computational particle numbers that exceed any realistically feasible value. While I am convinced that the general subject of the study is very important and that the study very well suits the journal Geoscientific Model Development, I am not (yet) convinced that the author's results are correct. In the following, I will suggest additional analyses required to accept this study for publication.

**Major Comments**

*Condensation-coalescence bottleneck.* Similar to the study by Hill et al. (2023), I believe that the high susceptibility to the number of computational particles is the very narrow droplet size distribution (DSD) caused by condensation, from which it is very hard to initiate collision-coalescence. For such distribution, producing sufficiently large precipitation embryos to start the collision-coalescence process is very important, and depends heavily on the number of computational particles to sample the tails of the DSD correctly. I suspect condensation narrowing is much more prominent in the kinematic simulations, in which condensation does not interact with the dynamics. Accordingly, the convergence in the kinematic simulations is much slower than in the dynamic simulation (Figs. 8 and 9). If this is true, the slow convergence cannot be seen as an inherent defect of Lagrangian cloud microphysical schemes, but must be considered a result of the artificially narrow DSD. To address this issue, additional analyses on the development of the SD width before the onset of collision-coalescence are highly recommended. Are there differences in the DSD width between the dynamic and kinematic simulations before collision-coalescence onset? How does the DSD width before collision-coalescence onset change with the computational particle number? If there are substantial differences between the dynamic and kinematic simulations, I highly recommend adding stochastic supersaturation fluctuations to the condensational growth [e.g., the approach by Grabowski and Abade (2017)] to obtain a realistic DSD width.

*More quantities.* The number of analyzed quantities to determine convergence is relatively low. While I understand the choice for the surface precipitation rate as the main subject, I recommend also checking the cloud base precipitation rate. Differences in the convergence of surface and cloud-base precipitation rates could indicate differences in the evaporation below cloud base. Furthermore, I miss deeper analyses of the convergence of the liquid water path (LPW) and rain water path (RWP), as well as the cloud droplet concentration. As all these parameters determine the surface precipitation rate, a comprehensive convergence on precipitation study should not negate them.

*Cloud-type dependence.* This study makes a strong statement on how many computational particles must be used in Lagrangian cloud microphysics schemes. However, these results are only obtained for one specific cloud type, a single cumulus congestus cloud. First, cumulus congestus are not a singularity. If a first cloud does not rain sufficiently, preconditioning might allow a second cloud to rain more, and vice versa. Thus, the convergence rate of an individual cloud has limited meaning. Thus, I recommend adding simulations of a cloud field. As I see that simulating several cumulus congestus can be cumbersome, I suggest simulating a standard shallow cumulus case (BOMEX, RICO). Furthermore, the convergence of drizzling stratocumulus can be a worthwhile extension (DYCOMS-II, RF02), primarily since they can easily be represented in a two-dimensional framework.

**Minor Comments**

L. 96: Is there a reference to support this statement?

L. 99: Give an equation for $P_{i,j}$.

Ll. 124 – 125: From what are the mean and standard deviation calculated? I assume the ensemble of simulations, but this should be stated clearly.

Ll. 139 – 140: How large is the ensemble?

Ll. 146 – 150: Where do we see this?

Ll. 160 – 162: How can one estimate the standard deviation from the SCE by taking the square root of the number of droplets? I think this approach can be correct, but the authors must elaborate.

Fig. 3, ll. 165 – 173: This analysis is distracting. I suggest removing it.

Ll. 203 – 206: If small differences result in differences in rain formation, they should be visible in the moments that are most susceptible to rain, e.g., the 6th moment of the DSD or the radar reflectivity, which is analyzed in Unterstrasser et al. (2017, 2020).

Eq. 3: What is "n"?

Fig. 5: Does the ordinate show "the number of simulations with P within a bin"? The resultant number of simulations seems to be very high.

L. 337: Stating that more than 1000 computational particles per grid box are necessary to simulate condensation correctly, needs to be supported by data.

Sec. 4.7: The initialization method might strongly impact how the aerosol size distribution is represented, and hence droplet activation. Thus, there might be differences in the droplet concentration and, commensurately, the rain rate.

Ll. 404 – 420: Can the approach by Schwenkel et al. (2018) help to accelerate convergence?

**Technical Comments**

The text is understandable, but there is a large number of spelling and grammatical errors that need to be taken care of.

L. 79: Write about "bin edges" already here. "Edges" without context is confusing.

Ll. 116 – 117, ff.: When narrative citations (\citet{…}) are used, a semicolon should not separate the individual references, but a comma or an "and".

**References**

Grabowski, W.W. and Abade, G.C., 2017. Broadening of cloud droplet spectra through eddy hopping: Turbulent adiabatic parcel simulations. *Journal of the Atmospheric Sciences*, *74*(5), pp.1485-1493.

Hill, A.A., Lebo, Z.J., Andrejczuk, M., Arabas, S., Dziekan, P., Field, P., Gettelman, A., Hoffmann, F., Pawlowska, H., Onishi, R. and Vié, B., 2023. Toward a numerical benchmark for warm rain processes. *Journal of the Atmospheric Sciences*.

Schwenkel, J., Hoffmann, F. and Raasch, S., 2018. Improving collisional growth in Lagrangian cloud models: development and verification of a new splitting algorithm. *Geoscientific Model Development*, *11*(9), pp.3929-3944.

Unterstrasser, S., Hoffmann, F. and Lerch, M., 2017. Collection/aggregation algorithms in Lagrangian cloud microphysical models: Rigorous evaluation in box model simulations. *Geoscientific Model Development*, *10*(4), pp.1521-1548.

Unterstrasser, S., Hoffmann, F. and Lerch, M., 2020. Collisional growth in a particle-based cloud microphysical model: insights from column model simulations using LCM1D (v1. 0). *Geoscientific Model Development*, *13*(11), pp.5119-5145.

---

## Referee Comment (RC2)

**Reviewer Comments to 'Modeling Collision-Coalescence in Particle Microphysics:.." by Zmijewski et al.**

As LCM has recently emerged as a next-generation cloud model, its rigorous assessment becomes very important. The present work carried out an important task of examining the numerical convergence, by extending the previous works such as Unterstrasser et al. (2017, 2020) and Dziekan and Pawlowska (2017). Simulations are carried out carefully, and the paper is generally well-written.

This paper provides valuable information to cloud modelers and is thus suitable for publication. On the other hand, the authors need to be more careful in drawing conclusions through deeper analysis, as the general validity of the conclusion is important for cloud modelers. The paper also lacks a proper explanation for the cause of the results. Most importantly, there is no clear explanation for why $\langle P \rangle$ and $\sigma$ do not converge with $N_{SD}^{bin}$ in 2D cumulus congestus (CC) simulations, contrary to the case of box simulations. Furthermore, the paper fails to elucidate why the decrease of $\langle P \rangle$ is caused by the increase of $\sigma$ in CC simulations. The hypothesis that 'A smoother spatial distribution of the DSD, together with mixing may lead to more precipitation' (L347) does not provide proper evidence (The convergence with $\Delta t_{coal}$ does not prove this hypothesis). It lacks an in-depth analysis of how DSD or $\sigma$ develops with time within a cloud in both dynamics and kinematic simulations. They also argued that the increased variance negatively affects precipitation without proper evidence (L350).

To clarify the difference between CC and box simulations, I suggest carrying out another set of simulations; that is, the multi-box simulations with mixing between boxes, possibly using isotropic turbulence simulation. It can bridge the gap between box and CC simulations and helps to clarify the effect of mixing between grids, as suggested by the authors. For example, one can examine how DSD and $\sigma$ are modified by directly comparing them with box simulation results. The direct comparisons of P and $\sigma$ are not possible between box and CC simulations. One of the most serious drawbacks of this paper is that there are no one-to-one simulation results for CC simulations, so one cannot assess how close the simulation with $N_{SD}^{bin} = 10^5$ is to reality. The multi-box simulations allow one-to-one simulations that can be used as the reference simulation. It also supports the generality of the present result from CC simulations.

*Other Comments,*

L40: a more detailed reference model ?

L125: There is no explanation of how the initial DSD is assigned for the LCM and SCE simulations, and how they are consistent with one-to-one simulations.

L162: I cannot understand why the authors suddenly discussed the LES model here.

L227: proportional to $N_{SD}^{bin-1}$? (L227)

L260: I do not agree that dimensionality does not affect numerical convergence. If mixing between grid boxes is important for numerical convergence, as argued by the authors, the number of neighboring grids is different between 2D and 3D.

L309: The difference in CWP is clear between LR, MR, and HR, contrary to the statement.

L348: Reference is required for 'lucky droplets'.

L365: only at large $N_{SD}^{bin}$

Fig. 2: σ does not converge with increasing $N_{SD}^{bin}$. What it shows is that σ approaches the result from one-to-one simulation. Without one-to-one simulation, the results are not that different from those from CC simulations (Fig. 9)

Fig. 2: I cannot see that $\langle m \rangle$ is smaller for $N_{SD}^{bin} = 10^2$ for $r > 50$ μm.

Fig. 5: The authors selected LR, MR, and HR based on $\langle P \rangle$, but Fig. 5 shows the frequency histogram of $P$.

Fig. 5: Horizontal axis is bin center? ($P/10^2$ ?)

Fig. 6: Isn't it natural to produce smoother variations, if the ensemble size of dynamical simulations is larger than that of kinematic simulations (= 20)? I also think that the authors should provide information on the ensemble size of box simulations and dynamic CC simulations (D) in the manuscript.

---

## Author Comment (AC1)

**Response to Reviewer #1**

We are grateful to the reviewer for comments. Please find our responses below. Reviewer's comments are in italics and our responses in normal style. Manuscript file with highlighted changes is available.

**1 Major Comments**

***Condensation-coalescence bottleneck.*** *Similar to the study by Hill et al. (2023), I believe that the high susceptibility to the number of computational particles is the very narrow droplet size distribution(DSD) caused by condensation, from which it is very hard to initiate collision-coalescence. For such distribution, producing sufficiently large precipitation embryos to start the collision-coalescence process is very important, and depends heavily on the number of computational particles to sample the tails of the DSD correctly. I suspect condensation narrowing is much more prominent in the kinematic simulations, in which condensation does not interact with the dynamics. Accordingly, the convergence in the kinematic simulations is much slower than in the dynamic simulation (Figs. 8 and 9). If this is true, the slow convergence cannot be seen as an inherent defect of Lagrangian cloud microphysical schemes, but must be considered a result of the artificially narrow DSD. To address this issue, additional analyses on the development of the SD width before the onset of collision-coalescence are highly recommended. Are there differences in the DSD width between the dynamic and kinematic simulations before collision-coalescence onset? How does the DSD width before collision-coalescence onset change with the computational particle number? If there are substantial differences between the dynamic and kinematic simulations, I highly recommend adding stochastic supersaturation fluctuations to the condensational growth [e.g., the approach by Grabowski and Abade (2017)] to obtain a realistic DSD width.*

We expanded the analysis of simulations without collision-coalescence by adding a figure with profiles of droplet number concentration, mean radius and relative dispersion. This is done to check how the DSD width before onset of collision-coalescence (which up

to that point is solely dependent on the condensational growth) changes with the number of SDs. We find that all profiles converge for 1000 SDs. That is faster than the convergence of precipitation, hence differences in precipitation are unlikely to be caused by differences in modeling of condensational growth.

The DSD width is similar in kinematic and dynamic simulations. Relative dispersion is around 0.2, indicating a narrow DSD, but within the range of observed values in stratiform (Miles et al., 2000; Pawlowska et al., 2006) and cumuliform (Lu et al., 2013) clouds. Therefore, the collision-coalescence algorithm should be expected to perform well for the DSD we get in the model. Nevertheless, we attempted to increase the DSD width by using the stochastic supersaturation model of Grabowski and Abade (2017) (henceforth GA17). Resulting profiles are shown in fig. 1. We observed a significant number of cloud droplets above the level at which there is cloud top in simulations without the GA17 model. This is most probably a consequence of randomness in GA17 – some 'lucky' SDs have a large positive supersaturation fluctuation, large enough for them to grow despite the fact that on average the grid cell is subsaturated. The number of such lucky SDs increases with the number of SDs, so we do not find convergence of results with the number of SDs. Due to these issues, we decided to not include simulations with GA17 in the paper.

***More quantities***. *The number of analyzed quantities to determine convergence is relatively low. While I understand the choice for the surface precipitation rate as the main subject, I recommend also checking the cloud base precipitation rate. Differences in the convergence of surface and cloud-base precipitation rates could indicate differences in the evaporation below cloud base. Furthermore, I miss deeper analyses of the convergence of the liquid water path (LPW) and rain water path (RWP), as well as the cloud droplet concentration. As all these parameters determine the surface precipitation rate, a comprehensive convergence on precipitation study should not negate them.*

Convergence of time series of cloud water path, RWP, cloud top height, cloud cover and surface precipitation is already shown in the supplement. In the main text, we have added plots of convergence of vertical profiles of precipitation flux. These help understand how differences in rain are distributed with height, including the cloud base level.

As described in the response to comment 1, we also added convergence plots for profiles of the liquid water content and cloud droplet concentration for simulations without collision-coalescence. These figures for simulations with collision-coalescence look very similar, hence we do not show them.

***Cloud-type dependence.*** *This study makes a strong statement on how many computational particles must be used in Lagrangian cloud microphysics schemes. However, these results are only obtained for one specific cloud type, a single cumulus congestus cloud. First, cumulus congestus are not a singularity. If a first cloud does not rain sufficiently, preconditioning might allow a second cloud to rain more, and vice versa. Thus, the convergence rate of an individual cloud has limited meaning. Thus, I recommend adding simulations of a cloud field. As I see that simulating several cumulus congestus can be cumbersome, I suggest simulating a standard shallow cumulus case (BOMEX, RICO). Furthermore, the convergence of drizzling stratocumulus can be a worthwhile extension (DYCOMS-II, RF02), primarily since they can easily be represented in a two-dimensional framework.*

The collision-coalsecence algorithm, which is the subject of this study, works in the same way in different cloud types. For generality, we study cases with different amounts of rain, but we do not agree that it is necessary to study different cloud types. It is true that for some cloud types, e.g. ones with a wider DSD, it may be easier to reach convergence in precipitation. However, we believe that it is better to study convergence for the more demanding, yet realistic cases (as in our study: the DSD is narrow, but within observational range). Convergence in the difficult case implies convergence in the simple case.

We consider modeling a single cloud as advantageous for studying collision-coalescence. It is true that if in a cloud field one cloud rains less than expected (e.g. due to errors in the collision-coalescence model), then there can be more water vapor left, or the aerosol size spectrum can be different due to in-cloud processing, and in consequence a subsequent cloud may produce more rain than expected. However, the collision-coalescence algorithm should correctly predict the amount of rain already in the first cloud and the effect described above would only make it more difficult to study correctness of the collision-coalescence algorithm.

**2   Minor comments**

- *L. 96: Is there a reference to support this statement?*

  Yes, we have added a reference to Shima et al. (2009).

- *L. 99: Give an equation for $P_{i,j}$.*

  It has been added.

- *Ll. 124 – 125: From what are the mean and standard deviation calculated? I assume the ensemble of simulations, but this should be stated clearly. Ll. 139 – 140: How*

*large is the ensemble?*

It is now clearly stated that these are statistics from an ensemble of simulations. The size of the ensemble is now given in Tab. 1.

- *Ll. 146 – 150: Where do we see this?*

  This can be seen in Fig.1, as now indicated in parentheses.

- *Ll. 160 – 162: How can one estimate the standard deviation from the SCE by taking the square root of the number of droplets? I think this approach can be correct, but the authors must elaborate.*

  This estimation, originating from Gillespie (1975), is introduced in section 3. We have added a reference to Gillespie (1975) in the lines in question.

- *Fig. 3, ll. 165 – 173: This analysis is distracting. I suggest removing it.*

  When comparing droplet size distributions on a log-log plot, as done in Figs. 1 and 2, it is easy to miss smaller errors. In the figure and lines in question, we plot the difference between the result and reference, which is a more detailed comparison. Differences revealed that way are consistent with differences in precipitation found in 2D simulations. For these reasons, we decide to keep this analysis in the manuscript.

- *Ll. 203 – 206: If small differences result in differences in rain formation, they should be visible in the moments that are most susceptible to rain, e.g., the 6th moment of the DSD or the radar reflectivity, which is analyzed in Unterstrasser et al. (2017, 2020).*

  We agree that higher moments are more sensitive to rain (large end of the DSD). Rain formation is in turn sensitive to collision-coalescence, so our intuition is that higher moments should converge more slowly. This is in fact what we find, contrary to Unterstrasser et al. (2017,2020). We have updated the paragraph in this spirit.

- *Eq. 3: What is "n"?*

  It is now clearly written that this is the ensemble size.

- *Fig. 5: Does the ordinate show "the number of simulations with P within a bin"? The resultant number of simulations seems to be very high.*

  Yes, it does. We conducted over 1000 dynamical simulations when generating velocity fields.

- *L. 337: Stating that more than 1000 computational particles per grid box are necessary to simulate condensation correctly, needs to be supported by data.*

  This statement was based on convergence of time series in simulations without collision-coalescence, which is discussed beforehand. Figures with profiles from these simulations, which have been added to the paper, further support this.

  The section about simulations without coalescence is now referenced in the line in question.

- *Sec. 4.7: The initialization method might strongly impact how the aerosol size distribution is represented, and hence droplet activation. Thus, there might be differences in the droplet concentration and, commensurately, the rain rate.*

  Initial size distribution for different initialization methods is shown for box simulations (Fig. 2). When averaged over multiple cells, the agreement is very good. This is now also stated in Sec. 4.7.

- *Ll. 404 – 420: Can the approach by Schwenkel et al. (2018) help to accelerate convergence?*

  Possibly it could help. We have added a comment about it in Conclusions.

**3   Technical Comments**

*The text is understandable, but there is a large number of spelling and grammatical errors that need to be taken care of.*

We have tried to fix such errors.

- *L. 79: Write about "bin edges" already here. "Edges" without context is confusing.*

  Changed to "bin edges".

- *Ll. 116 – 117, ff.: When narrative citations ($\cite\{...\}$) are used, a semicolon should not separate the individual references, but a comma or an "and".*

  Style of citations is defined by the GMD Latex template.

[Figure]

Figure 1. Profiles of cloud droplet concentration (top row), cloud droplet mean radius (center row), and relative dispersion of cloud droplet radius (bottom row) from HR simulations with GA17 and without collision-coalescence. These profiles are averaged across cloudy cells and over a time interval ranging from 1800 s to 9600 s within the ensemble of simulations. Columns show results for different values of the TKE dissipation rate (different SGS turbulence strength): $0\,\mathrm{cm^2s^{-3}}$ (left), $1\,\mathrm{cm^2s^{-3}}$ (center) and $20\,\mathrm{cm^2s^{-3}}$ (right).

**References**

Gillespie, D. T.: Exact Method for Numerically Simulating the Stochastic Coalescence Process in a Cloud., Journal of the Atmospheric Sciences, 32, 1977–1989, https://doi.org/10.1175/1520-0469(1975)032¡1977:AEMFNS¿2.0.CO;2, 1975.

Lu, C., Niu, S., Liu, Y., and Vogelmann, A. M.: Empirical relationship between entrainment rate and microphysics in cumulus clouds, Geophysical Research Letters, 40, 2333–2338, https://doi.org/https://doi.org/10.1002/grl.50445, 2013.

Miles, N. L., Verlinde, J., and Clothiaux, E. E.: Cloud Droplet Size Distributions in Low-Level Stratiform Clouds, Journal of the Atmospheric Sciences, 57, 295 – 311, https://doi.org/https://doi.org/10.1175/1520-0469(2000)057¡0295:CDSDIL¿2.0.CO;2, 2000.

Pawlowska, H., Grabowski, W. W., and Brenguier, J.-L.: Observations of the width of cloud droplet spectra in stratocumulus, Geophysical Research Letters, 33, https://doi.org/https://doi.org/10.1029/2006GL026841, 2006.

Shima, S., Kusano, K., Kawano, A., Sugiyama, T., and Kawahara, S.: The super-droplet method for the numerical simulation of clouds and precipitation: A particle-based and probabilistic microphysics model coupled with a non-hydrostatic model, Quarterly Journal of the Royal Meteorological Society, 135, 1307–1320, https://doi.org/10.1002/qj.441, 2009.

---

## Author Comment (AC2)

**Response to Reviewer #2**

We are grateful to the reviewer for comments. Please find our responses below. Comments are in italics and our responses in normal style. Manuscript file with highlighted changes is available.

**1   Major Comments**

*Validity of the conclusion This paper provides valuable information to cloud modelers and is thus suitable for publication. On the other hand, the authors need to be more careful in drawing conclusions through deeper analysis, as the general validity of the conclusion is important for cloud modelers. The paper also lacks a proper explanation for the cause of the results. Most importantly, there is no clear explanation for why $\langle P \rangle$ and $\sigma$ do not converge with $N_{SD}^{bin}$ in 2D cumulus congestus (CC) simulations, contrary to the case of box simulations. Furthermore, the paper fails to elucidate why the decrease of $\langle P \rangle$ is caused by the increase of $\sigma$ in CC simulations. The hypothesis that 'A smoother spatial distribution of the DSD, together with mixing may lead to more precipitation' (L347) does not provide proper evidence (The convergence with $\Delta t_{coal}$ does not prove this hypothesis). It lacks an in-depth analysis of how DSD or $\sigma$ develops with time within a cloud in both dynamics and kinematic simulations. They also argued that the increased variance negatively affects precipitation without proper evidence (L350).*

It is clear why $\sigma$ does not converge in CC simulations (nor in box simulations). That is because $N_{SD}^{bin}$ is smaller than the number of real droplets, hence the number of random trials for collision is smaller than it should be (Shima et al., 2009).

We do not claim to know why $\langle P \rangle$ converges slower in CC than in box. We do hypothesize that it is linked to the spatial distribution of the DSD. We made changes in the text to emphasize that this is a hypothesis and not a conclusion. To give the hypothesis more substance, we added a figure with a frequency histogram of the rain water content in the cloud at different moments in time. It shows that for small $N_{SD}^{bin}$ the spatial distribution

of rain is less uniform than for large $N_{SD}^{bin}$ and that accumulated precipitation converges when its spatial distribution converges. We do not agree that convergence with $\Delta t_{coal}$ is not compatible with this hypothesis. Box simulations show that $\sigma$ is not sensitive to $\Delta t_{coal}$. The highly speculative paragraph at L350 has been removed.

*Difference between CC and box simulations To clarify the difference between CC and box simulations, I suggest carrying out another set of simulations; that is, the multi-box simulations with mixing between boxes, possibly using isotropic turbulence simulation. It can bridge the gap between box and CC simulations and helps to clarify the effect of mixing between grids, as suggested by the authors. For example, one can examine how DSD and $\sigma$ are modified by directly comparing them with box simulation results. The direct comparisons of P and $\sigma$ are not possible between box and CC simulations. One of the most serious drawbacks of this paper is that there are no one-to-one simulation results for CC simulations, so one cannot assess how close the simulation with $N_{SD}^{bin} = 10^5$ is to reality. The multi-box simulations allow one-to-one simulations that can be used as the reference simulation. It also supports the generality of the present result from CC simulations.*

We performed multi-box simulations as suggested. Their results are discussed in a new section after the box simulations section. Simulated domain is the same as in box simulations, but it is divided into smaller cells. SDs move due to isotropic turbulence and sedimentation. We found that in one-to-one simulations the DSD in multi-box is the same as in box even for very small multi-box cells that on average contain only one real droplet. When SDs represent more than one real droplet, we observe discrepancies from the reference result as cell size is decreased. Nevertheless, fewer SDs per cell are needed in multi-box than in box simulations. This suggests that mixing helps reach convergence, as already observed by Schwenkel et al. (2018). To understand how mixing affects convergence in CC, we also ran CC simulations with a model for sub-grid scale motion of super-droplets. We found that using this model helps reach convergence in precipitation, hence mixing plays a positive role in CC just like it does in multi-box simulations.

**2 Minor comments**

- *L40: a more detailed reference model ?*

    We now explicitly mention the name of the more detailed model, which is the one-to-one simulation, and provide a reference to Dziekan and Pawlowska (2017).

- *L125: There is no explanation of how the initial DSD is assigned for the LCM and*

*SCE simulations, and how they are consistent with one-to-one simulations.*

The method for initializing SD radii and multiplicities from a prescribed DSD is detailed in subsection 2.1 titled 'Initialization of SD Radii and Multiplicities'. SCE simulations are solved with the well-known flux method Bott (1997). To make sure that the initial DSD is the same for all methods used, we show it in Fig. 2 (a).

- *L162: I cannot understand why the authors suddenly discussed the LES model here.*

  Based on your suggestion, we have removed the discussion about the LES model. It indeed seemed out of place and could be distracting.

- *L227: proportional to $N_{SD}^{bin-1}$? (L227)*

  Yes, we've fixed it.

- *L260: I do not agree that dimensionality does not affect numerical convergence. If mixing between grid boxes is important for numerical convergence, as argued by the authors, the number of neighboring grids is different between 2D and 3D.*

  We agree, the paragraph has been revised.

- *L309: The difference in CWP is clear between LR, MR, and HR, contrary to the statement*

  The statement has been revised.

- *L348: Reference is required for 'lucky droplets'.*

  The paragraph about 'lucky droplets' has been removed as it was too speculative.

- *L365: only at large $N_{SD}^{bin}$*

  We don't agree. Even at small $N_{SD}^{bin}$, differences in the flow field (D simulations) give 4 times greater relative standard deviation of precipitation than differences in realization of collision-coalescence (MR simulations).

- *Fig. 2: $\sigma$ does not converge with increasing $N_{SD}^{bin}$. What it shows is that $\sigma$ approaches the result from one-to-one simulation. Without one-to-one simulation, the results are not that different from those from CC simulations (Fig. 9)*

  Yes, we agree. That's what we were trying to convey in the paper.

- *Fig. 2: I cannot see that $\langle m \rangle$ is smaller for $N_{SD}^{bin}$ $10^2$ for $r > 50\mu m$.*

  This can't be seen in Fig. 2 because of the logarithmic axis. It is visible in Fig. 3 (d). Figure 3 (d) is now referenced in the line where we write that $\langle m \rangle$ is smaller for $N_{SD}^{bin}$ $10^2$ for r > $50\mu m$.

- *Fig. 5: The authors selected LR, MR, and HR based on ⟨P⟩, but Fig. 5 shows the frequency histogram of P.*

  Correct. To show a frequency histogram of ⟨P⟩ (average from an ensemble of kinematic simulations) we would need to run an ensemble of simulations for velocity field from each of around 1000 dynamical simulations shown in Fig. 5. That would be massive work for no clear reason, since our goal was to find 3 velocity fields that give different ⟨P⟩. Once such 3 fields were found, we stopped running kinematic ensembles.

- *Fig. 5: Horizontal axis is bin center? (P/10$^2$ ?)*

  Correct. The values are small because that is accumulated precipitation over the entire simulation time and the entire simulation domain, while the cloud precipitates only for a short time and over a small part of the domain. Maximum of the domain-averaged precipitation rate divided by cloud cover is around 10 mm/h (HR case).

- *Fig. 6: Isn't it natural to produce smoother variations, if the ensemble size of dynamical simulations is larger than that of kinematic simulations (= 20)? I also think that the authors should provide information on the ensemble size of box simulations and dynamic CC simulations (D) in the manuscript.*

  The final simulation ensemble of kinematic simulations for $N_{SD}^{bin} = 10^2$ (that is shown in the figure in question) was much larger than 20. We modified the text to emphasize that these 20 simulations were just a preliminary ensemble. The final ensemble size is now given in Tab.1 (for CC simulations) and in the text (for box and multi-box simulations).

**References**

Bott, A.: An efficient numerical flux method for the solution of the stochastic collection equation, Journal of Aerosol Science, 28, 2284–2293, https://doi.org/10.1016/S0021-8502(97)85371-2, 1997.

Dziekan, P. and Pawlowska, H.: Stochastic coalescence in Lagrangian cloud microphysics, Atmospheric Chemistry and Physics, 17, 13 509–13 520, https://doi.org/10.5194/acp-17-13509-2017, 2017.

Schwenkel, J., Hoffmann, F., and Raasch, S.: Improving collisional growth in Lagrangian cloud models: development and verification of a new splitting algorithm, Geoscientific Model Development, 11, 3929–3944, https://doi.org/10.5194/gmd-11-3929-2018, 2018.

Shima, S., Kusano, K., Kawano, A., Sugiyama, T., and Kawahara, S.: The super-droplet method for the numerical simulation of clouds and precipitation: A particle-based and probabilistic microphysics model coupled with a non-hydrostatic model, Quarterly Journal of the Royal Meteorological Society, 135, 1307–1320, https://doi.org/10.1002/qj.441, 2009.

---

## Author Comment (AC3)

**Response to Reviewer #1**

We are grateful to the reviewer for comments. Please find our responses below. Reviewer's comments are in italics and our responses in normal style. Manuscript file with highlighted changes is available.

**1 Minor Comments**

*Ll. 1 – 17, 515 – 530: Explicitly state how much subgrid-scale fluctuations accelerate convergence.*

Thanks to a suggestion from the Reviewer 2, we realized that SGS motion causes depletion of aerosols and super-droplets near the surface. This caused a decrease in cloud droplet concentration, what resulted in an increase in precipitation. We did a new set of simulations with SGS motion and with aerosol relaxation that counters the depletion. Results are in agreement with simulations without SGS motion. Therefore, we no longer believe that SGS motion helps accelerate convergence.

*Ll. 228 – 231 and Sec. 2.1: How well do the initialization method capture the large tail of the droplet size distribution? The large tail is most important for the initialization of precipitation, and hence the higher-order moments of the droplet size distribution. A figure showing higher moments of the initial droplet size distribution for different numbers of simulated particles would reveal if there is a dependency on the initial conditions. I suspect these higher moments are not converged, so the subsequent simulations struggle to converge. All moments of the initial droplet size distribution should agree for a fair comparison.*

In the Supplement, we added a plot of the first 11 moments of the initial distribution in box simulations. These are representative also of the 2D cloud simulations, because they use the same initialization method. Moments agree very well between simulations

with different numbers of super-droplets, so this does not explain differences in results for different number of SDs.

*Ll. 295 − 296: The reference to Grabowski and Abade (2017) is misleading. First, their paper is about a subgrid-scale model for supersaturation fluctuations, and does not primarily focus on velocity fluctuations. Second, subgrid-scale models for velocity fluctuations in Lagrangian models exist for much longer (e.g., Weil et al. 2004). Third, how is the subgrid-scale model coupled? Does it obtain some information on subgrid-scale turbulence kinetic energy?*

We did not intend to say that the model was devised in Grabowski and Abade (2017). Various types of the Ornstein-Uhlenbeck processe had been used before to model turbulence. We give this reference, because it contains the exact equation we use in our simulation. For example, the Weil et al. (2004) model is different. Now, we give the number of equation in Grabowski and Abade (2017) that is relevant, and we no longer call the model GA17, but OU (for Ornstein-Uhlenbeck).

**2   Technical comments**

*I repeat my previous comment: "When narrative citations (**?**) are used, a semicolon should not separate the individual references, but a comma or an 'and'." The authors claim that the GMD LaTeX template allows this, but it causes grammatically wrong sentences. For instance, instead of writing "In line with conclusions of Schwenkel et al. (2018); Unterstrasser et al. (2020), multi-box simulations show [...]" the authors should write "In line with conclusions of Schwenkel et al. (2018) and Unterstrasser et al. (2020), multi-box simulations show [...]". The semicolon separates the sentence in two meaningless parts. Only because one can create such citations with the template, they are not correct!*

We have fixed narrative citations.

---

## Author Comment (AC4)

**Response to Reviewer #2**

We are grateful to the reviewer for comments. Please find our responses below. Reviewer's comments are in italics and our responses in normal style. Manuscript file with highlighted changes is available.

**1   Major Comments**

*It is not clear how the multi-box simulation results can explain the CC results from the perspective of intercell mixing. Intercell mixing helps multi-box simulations to reach the convergence of mean DSD for smaller $N_{\text{SD}}^{\text{(bin)}}$ than in box simulations (L250, L475), but it prevents the mean precipitation in the CC simulation from reaching convergence, especially in the case of weak precipitation.*

We do not claim that intercell mixing prevents convergence in CC simulations, and we do not know why convergence is slower in CC than in box/multi-box. What we find is that mean precipitation converges when the spatial distribution of rain water converges. This might be pure coincidence, but it may also mean that mean precipitation depends on the spatial distribution of rain. If it does, it has to be due to intercell interactions. The following paragraph has been added to conclusions to convey these ideas more clearly:

" It is not clear why convergence is slower in CC than in box simulations. In CC, convergence of mean precipitation coincides with convergence of the spatial distribution of rain. This may suggest that mean precipitation is dependent on spatial distribution of droplet sizes, probably because of interaction between cells. However, in multi-box simulations we observe that intercell mixing helps reach convergence. Increasing the rate of intercell mixing in CC by using a SGS model does not help with convergence. This does not necessarily indicate that intercell mixing is not important for convergence in CC. It is possible that the increase in intercell mixing caused by the SGS model is small in relation to interecell mixing caused by resolved eddies and by sedimentation. Precipitation is sensitive to the super-droplet initialization procedure. In this study initial radii were

almost evenly distributed on a logarithmic scale. If droplet radii are randomly drawn from the initial distribution, it is more difficult to reach convergence with $N_{\text{SD}}^{(\text{bin})}$ and using too small $N_{\text{SD}}^{(\text{bin})}$ induces larger errors. "

*The inclusion of SGS motions of SDs enhances the mean precipitation greatly. The authors attributed this to enhanced intercell mixing. It is difficult to imagine that the enhancement of intercell mixing is generated greatly by including SGS motions of SDs since intercell mixing occurs mainly by resolved eddies and sedimentation. The strong sensitivity of SGS motions of SDs to precipitation requires a more in-depth analysis, because it has important implications in cloud models. Probably the authors need to investigate the modification of DSD and intercell mixing by SGS motions of SDs.*

We did a deeper analysis of simulations with SGS motion of SDs and it revealed that SGS motion leads to a depletion of SDs (and aerosols that these SDs represent) near the surface. Random SGS velocities cause SDs to hit the surface. Depletion of aerosols results in fewer cloud droplets and more precipitation. This, and not enhanced intercell mixing, was the main reason why there was more precipitation in simulations with SGS motion. In general, simulations with more precipitation converge more easily. This (and not necessarily a positive role of SGS motion) may explain better convergence of simulations with SGS motion.

We made a new ensemble of simulations with SGS motion, in which we add SDs near the surface to counter the depletion of aerosols. Parameters of this relaxation procedure have been tuned to obtain the same aerosol concentration and number of SDs as in simulations without SGS motion. Precipitation in these new simulations is almost the same as in simulations without SGS motion. It does not necessarily mean that intercell mixing does not help with convergence in CC simulations. It is possible that it does help, but the rate of intercell mixing due to SGS motion is small compared to that caused by resolved flow and sedimentation.

*I think the authors should make clear in conclusions that the CC simulation results in the dynamical simulation with the SGS motion on SDs (Fig. 15) are less affected by the non- convergence (Fig. 11).*

A new set of simulations with SGS motion showed that SGS motion does not help much with convergence (see answer to the previous comment).

**2  Minor Comments**

*1. The term 'mixing' is confusing to me; that is, mixing within a cell, which helps to produce uniform DSD within a cell, and intercell mixing.*

Wherever ambiguous, we have replaced 'mixing' with 'intercell mixing'. Simulations with a model for subgrid-scale motion of SDs are now labeled 'SGS SD motion' instead of 'mixing'.

*2. I hope the authors select the line color more systematically in Fig. 8, 9, and 12; i.e., from blue to red with increasing $N_{\text{SD}}^{(\text{bin})}$.*

Colors now change gradually as in a sequential matplotlib colormap 'copper'. Color is proportional to the logarithm of $N_{\text{SD}}^{(\text{bin})}$.

*3. L395; I do not think they are consistent. P increases monotonically in the box simulations (Fig. 3). I also cannot understand how the consistency can be explained by the convergence of $N_{\text{SD}}^{(\text{bin})}$.*

We extend this paragraph to make it more clear why we think box and CC simulations are consistent for $N_{\text{SD}}^{(\text{bin})} \leq 10^3$:

"Changes of $\langle P \rangle$ for $N_{\text{SD}}^{(\text{bin})} \leq 10^3$ are consistent with the changes in mean DSD in box simulations (see Fig. 3 (d)). In box simulations, $N_{\text{SD}}^{(\text{bin})} = 10$ gives too few droplets with radii between 40 and 120 microns, but too many droplets with radii greater than $120\,\mu\text{m}$. Since surface precipitation is sensitive to the largest droplets, this is consistent with too large $\langle P \rangle$ seen in CC simulations. For $N_{\text{SD}}^{(\text{bin})} = 10^2$, number of the largest droplets is no longer overestimated in box simulations, but there are still too few droplets with radii between 50 and 120 microns. This is consistent with a sharp decrease of $\langle P \rangle$ between $N_{\text{SD}}^{(\text{bin})} = 10$ and $N_{\text{SD}}^{(\text{bin})} = 10^2$. In box simulations with $N_{\text{SD}}^{(\text{bin})} = 10^3$ the number of droplets with sizes between 50 and 120 microns is no longer underestimated, what is consistent with an increase of $\langle P \rangle$ between $N_{\text{SD}}^{(\text{bin})} = 10^2$ and $N_{\text{SD}}^{(\text{bin})} = 10^3$ in CC simulations."

---

## Author Response (AR1)

**According to the message:**

*Justification (visible to authors and reviewers only):*
*Dear Authors,*

*the manuscript is well-written and is well within the scope of GMD.*

*It is GMD guideline that the name and version of the employed model should appear in the title. (personally, I could live with your title, but we cannot ignore this requirement)*

*Best wishes, Simon*

**We modified the manuscript title into:**

*Modeling Collision-Coalescence in Particle Microphysics: Numerical Convergence of Mean and Variance of Precipitation in Cloud Simulations Using University of Warsaw Lagrangian Cloud Model (UWLCM) 2.1*

**Manuscript revision into new version**

We have diligently implemented the reviewers' suggestions into the manuscript, making sure all changes are clearly visible in the Author's track-changes file. These revisions are in line with our responses to the reviewers' comments. Additionally, in each of our responses, we've provided a detailed record of the specific modifications made, following this format: reviewer's question, our response, and corresponding manuscript changes.

---

## Referee Report (RR1)

**Review of "Modeling Collision-Coalescence in Particle Microphysics: Numerical Convergence of Mean and Variance of Precipitation in Cloud Simulations Using University of Warsaw Lagrangian Cloud Model (UWLCM) 2.1" by Zmijewski et al. (gmd-2023-44)**

The response letter and the revised manuscript address most of my concerns satisfactorily. I especially enjoyed the addition of the multi-box simulations and the additional analyses on the effects of subgrid-scale velocity fluctuations on the convergence. While I have some minor comments below, I consider this manuscript almost ready for publication.

**Minor Comments** (line numbers refer to the tracked-changes document)

Ll. 1 – 17, 515 – 530: Explicitly state how much subgrid-scale fluctuations accelerate convergence.

Ll. 228 – 231 and Sec. 2.1: How well do the initialization method capture the large tail of the droplet size distribution? The large tail is most important for the initialization of precipitation, and hence the higher-order moments of the droplet size distribution. A figure showing higher moments of the initial droplet size distribution for different numbers of simulated particles would reveal if there is a dependency on the initial conditions. I suspect these higher moments are not converged, so the subsequent simulations struggle to converge. All moments of the initial droplet size distribution should agree for a fair comparison.

Ll. 295 – 296: The reference to Grabowski and Abade (2017) is misleading. First, their paper is about a subgrid-scale model for supersaturation fluctuations, and does not primarily focus on velocity fluctuations. Second, subgrid-scale models for velocity fluctuations in Lagrangian models exist for much longer (e.g., Weil et al. 2004). Third, how is the subgrid-scale model coupled? Does it obtain some information on subgrid-scale turbulence kinetic energy?

**Technical Comments**

I repeat my previous comment: "When narrative citations (\citet{…}) are used, a semicolon should not separate the individual references, but a comma or an 'and'." The authors claim that the GMD LaTeX template allows this, but it causes grammatically wrong sentences. For instance, instead of writing "In line with conclusions of Schwenkel et al. (2018); Unterstrasser et al. (2020), multi-box simulations show […]" the authors should write "In line with conclusions of Schwenkel et al. (2018) and Unterstrasser et al. (2020), multi-box simulations show […]". The semicolon separates the sentence in two meaningless parts. Only because one can create such citations with the template, they are not correct!

**References**

Grabowski, W.W. and Abade, G.C., 2017. Broadening of cloud droplet spectra through eddy hopping: Turbulent adiabatic parcel simulations. *Journal of the Atmospheric Sciences*, *74*(5), pp.1485-1493.

Weil, J.C., Sullivan, P.P. and Moeng, C.H., 2004. The use of large-eddy simulations in Lagrangian particle dispersion models. *Journal of the Atmospheric Sciences*, *61*(23), pp.2877-2887.

---

## Referee Report (RR2)

**2nd Reviewer Comments to 'Modeling Collision-Coalescence in Particle Microphysics:.."
by Zmijewski et al.**

The authors included the analyses of the results from multi-box simulation results and simulations with SGS motions of SDs, which made the paper more valuable. On the other hand, analyses and interpretations of the newly added simulations appear to be insufficient to contribute to a better understanding of the overall results.

It is not clear how the multi-box simulation results can explain the CC results from the perspective of intercell mixing. Intercell mixing helps multi-box simulations to reach the convergence of mean DSD for smaller $N_{\mathrm{SD}}^{(\mathrm{bin})}$ than in box simulations (L250, L475), but it prevents the mean precipitation in the CC simulation from reaching convergence, especially in the case of weak precipitation.

The inclusion of SGS motions of SDs enhances the mean precipitation greatly. The authors attributed this to enhanced intercell mixing. It is difficult to imagine that the enhancement of intercell mixing is generated greatly by including SGS motions of SDs since intercell mixing occurs mainly by resolved eddies and sedimentation. The strong sensitivity of SGS motions of SDs to precipitation requires a more in-depth analysis, because it has important implications in cloud models. Probably the authors need to investigate the modification of DSD and intercell mixing by SGS motions of SDs.

I think the authors should make clear in conclusions that the CC simulation results in the dynamical simulation with the SGS motion on SDs (Fig. 15) are less affected by the non-convergence (Fig. 11).

I consider that the present paper will provide important information on cloud modelers, once the above ambiguities are clarified.

Minor:

1. The term 'mixing' is confusing to me; that is, mixing within a cell, which helps to produce uniform DSD within a cell, and intercell mixing.

2. I hope the authors select the line color more systematically in Fig. 8, 9, and 12; i.e., from blue to red with increasing $N_{\mathrm{SD}}^{(\mathrm{bin})}$.

3. L395; I do not think they are consistent. P increases monotonically in the box simulations (Fig. 3). I also cannot understand how the consistency can be explained by the convergence of $N_{\text{SD}}^{(\text{bin})}$.

---

## Editor Decision (ED1)

[revised manuscript text omitted]
_{\text{SD}}^{\text{(bin)}}$: $N_{\text{SD}}^{\text{(cell)}}$ can be smaller for larger $N_{\text{SD}}^{\text{(bin)}}$ (e.g. $N_{\text{SD}}^{\text{(cell)}} = 10$ works well for $N_{\text{SD}}^{\text{(bin)}} = 10^4$, but $N_{\text{SD}}^{\text{(cell)}} = 12.5$ gives errors for $N_{\text{SD}}^{\text{(bin)}} = 10^2$). The number of
250 coalescence cells does not affect $\sigma(m)$ (not shown in the figure).

In line with conclusions of Schwenkel et al. (2018) and of Unterstrasser et al. (2020), multi-box simulations show that fewer SDs per cell are needed to correctly model collision-coalescence when mixing of SDs between cells is included. For example, box simulations with $N_{\text{SD}}^{\text{(bin)}} = N_{\text{SD}}^{\text{(cell)}} = 10$ give significant errors, but multi-box simulations with $N_{\text{SD}}^{\text{(bin)}} = 10^4$ and $N_{\text{SD}}^{\text{(cell)}} = 10$ are close to reference. It is important that the rate of intercell mixing decreases with increasing cell size,
255 what affects the minimal required $N_{\text{SD}}^{\text{(cell)}}$. Simulations for $N_{\text{SD}}^{\text{(bin)}} = 10^2$ and $N_{\text{SD}}^{\text{(cell)}} = 12.5$ give errors, while simulations for $N_{\text{SD}}^{\text{(bin)}} = 10^4$ and $N_{\text{SD}}^{\text{(
[revised manuscript text omitted]